# Design of hierarchical-heterostructure antiferroelectrics for ultrahigh capacitive energy storage

Liang Chen [1,5], Tengfei Hu [2,5], He Qi[3] ✉, Huifen Yu[1], Zhengqian Fu [2], Shujun Zhang [4] & Jun Chen [1,3] ✉

Electrostatic dielectric capacitors with high power density are the fundamental energy storage components in advanced electronic and electric power systems. However, simultaneously achieving ultrahigh energy density and efficiency poses a persistent challenge, preventing the capacitive applications towards miniaturization and low-energy consumption. Here we demonstrate giant energy storage properties in lead-free antiferroelectrics by designing hierarchical heterostructures to optimize polarization evolution paths. Through the design of antiferroelectric nanoclusters featuring interlocked polarization structure and fishbone polarization configuration, alongside order-disorder oxygen octahedral tilts, we increase polarization fluctuation and delay polarization saturation with nearly eliminated hysteresis under ultrahigh external electric fields. Leveraging this strategy, we achieve an ultrahigh energy density of 21.0 J cm⁻³ with an impressive efficiency of 90% in sodium niobate-based ceramics, underscoring the great potential of this methodology for designing high-performance dielectrics and other functional materials.

Dielectric capacitors are extensively used in advanced electronic and electric power systems, benefiting from their superiority of high power density $P_D$, ultrafast charge-discharge rate $t_{0.9}$, and excellent energy storage stability[1-4]. From a practical application standpoint, compared with thin film capacitors, ceramic capacitors offer distinct advantages, including simple fabrication, low cost, high power and energy storage capacity, thus attracting growing attention in the field of dielectric energy storage[5,6]. The low energy density $U_e$ and suboptimal efficiency $\eta$, however, largely limit the long-term development of ceramic capacitors towards miniaturization and low-energy consumption[5,7,8]. Consequently, extensive efforts have been dedicated to develop ceramic capacitors with outstanding comprehensive energy storage capabilities[4,9].

According to the theory of electrostatic energy storage, $U_e$ and $\eta$ are determined by the polarization $P$ response of dielectrics under an external electric field $E$, that are, $U_e = \int_{P_r}^{P_m} E dP$ and $\eta = U_e/(U_e + U_{loss})$, where $P_m$, $P_r$, and $U_{loss}$ represent the maximum polarization, remnant polarization, and energy loss, respectively (Supplementary Fig. S1)[10]. Therefore, the large polarization fluctuation ($\Delta P = P_m - P_r$, large $P_m$ and small $P_r$) and high breakdown electric field $E_b$ should be effectively controlled to achieve high $U_e$ and $\eta$, simultaneously[11]. Over the past few years, $E_b$ has been greatly enhanced by various approaches, such as improving band gaps, decreasing strain and refining grains, minimizing voids, and mitigating energy loss from the perspectives of intrinsic, electromechanical, partial discharge, and thermal breakdown,

¹Beijing Advanced Innovation Center for Materials Genome Engineering, Department of Physical Chemistry, University of Science and Technology Beijing, Beijing, China. ²State Key Laboratory of High Performance Ceramics & The Key Lab of Inorganic Functional Materials and Devices, Shanghai Institute of Ceramics, Chinese Academy of Sciences, Shanghai, China. ³State Key Laboratory of Tropic Ocean Engineering Materials and Materials Evaluation, Hainan University, Haikou, Hainan Province, China. ⁴Institute for Superconducting and Electronic Materials, Australian Institute of Innovative Materials, University of Wollongong, Wollongong, NSW, Australia. ⁵These authors contributed equally: Liang Chen 陈良, Tengfei Hu 胡腾飞. ✉e-mail: qihe@hainanu.edu.cn; junchen@ustb.edu.cn

respectively, reaching a level up to 80–100 kV mm$^{-15,12-14}$. For controlling the desired polarization fluctuation under external electric fields, on the other hand, the strategies including domain engineering, polymorphic polar nanoregions (PNRs), multistage phase transition have been employed to obtain a large $\Delta P$ of 50–60 μC cm$^{-2[14-17]}$. It is noteworthy that both $E_b$ and $\Delta P$ have nearly approached the upper limits observed in perovskite ceramics prepared by traditional solid-solution method, leading to the unilateral achievement of ultrahigh $U_e$ (10-15 J cm$^{-3}$). However, the great challenge of realizing ultrahigh energy storage density with simultaneous ultrahigh efficiency still persists in ceramic capacitors.

To overcome the bottleneck in achieving comprehensive energy storage properties, optimizing polarization evolution paths from the perspective of local structure emerges as the critical avenue, particularly at high $E_b$ and large $\Delta P$ values. Polarization and oxygen octahedral ($BO_6$) tilt, as two types of local structural units that can be excited by external electric fields, directly and significantly govern the macroscopic polarization response process of dielectrics induced by electric fields. It is recognized that antiferroelectric supercells are coupled by antiparallel polarization and $BO_6$ tilt, offering substantial potential for local structural regulation to realize outstanding energy storage properties[18,19]. Considering the recent research progresses, however, the regulation of local polarization structure has always been focused on (relaxor) ferroelectrics, lacking exploration and understanding of internal polarization in (relaxor) antiferroelectrics, let alone the meticulous design and synthesis of novel local antiferroelectric polarization landscapes. $BO_6$ tilt can cause severe polarization hysteresis, but it can effectively delay polarization saturation[4,20]. The control of $BO_6$ tilt configuration is often overlooked in dielectric energy storage, losing another key way to suppress the rapid saturation behavior of relaxor ferroelectrics and solve the large polarization hysteresis of relaxor antiferroelectrics, which have always been a major challenge in designing high-performance dielectrics, especially lead-free antiferroelectrics. If the polarization and $BO_6$ tilt can be synergistically controlled to ingeniously construct diverse local heterostructures, polarization hysteresis is expected to be significantly suppressed while delaying polarization saturation and maintaining a large $\Delta P$, developing high-performance dielectrics for next-generation advanced energy storage applications.

Here, we propose a hierarchical heterostructure design strategy that controls favorable polarization evolution through the synergistic regulation of polarization and $BO_6$ tilt, aiming to enhance energy storage properties (Fig. 1)[21]. High-bandgap NaNbO$_3$ (NN, tolerance factor: $t = 0.967$) ceramic with antiferroelectric nature is selected as the implementation target, in which antiparallel off-centering cation displacements in adjacent unit cells and $BO_6$ distortions existed[22,23]. To preserve the energy storage advantages of antiferroelectrics, low-$t$ (<1) modifiers are chosen to stabilize antiferroelectric order. Considering the solubility limitations, 10 mol% of the low-$t$ (0.914) and nonpolar CaZrO$_3$ (CZ) is first doped into NN to form an NN-CZ antiferroelectric matrix. This can decrease polarization displacement and enhance component disorder as much as possible to break long-range antiferroelectric polarization order, transforming the concave polarization rising path into a convex one[20,24,25]. However, the ordered $BO_6$ distortion with large tilt angles can be obtained by doping CZ, which can deteriorate $P_m$ and $\eta$ despite effectively delaying polarization saturation[20]. To ensure polarization evolution with negligible hysteresis and a large $\Delta P$, we introduce large-$t$ (0.975) and high-spontaneous polarization ($P_s$) Bi$_{0.5}$Na$_{0.5}$TiO$_3$ (BNT) into NN-CZ matrix to synthesize (1-$x$)(NN-CZ)-$x$BNT (NN-CZ-$x$BNT; $x = 0$ to 0.25) solid solutions to further localize polarization, enhance local polar distortion, and weaken $BO_6$ order[7,26,27]. Such designs are desired to synthesize hierarchical heterostructures with diverse antiferroelectric nanoclusters and order-disorder $BO_6$ tilts, thereby improving the comprehensive energy storage performance of the dielectrics.

## Results and discussion
### General structure features

We first demonstrate that all the ceramics are pure perovskite structure without any secondary phase through X-ray and neutron diffraction patterns (Supplementary Fig. S2). Both in-phase and anti-phase $BO_6$ tilts are further identified in all dielectrics, evidenced by the appearance of three superlattice peaks corresponding to ($ooe$)/2 and ($ooo$)/2 ($o$ is odd and $e$ is even) based on the conclusions by *Glazer*[20,23,26,28,29]. The weakened superlattice peaks with increasing BNT means the abatement of $BO_6$ tilts and antiferroelectric nature, which can also be confirmed by the gradually flattening and decreasing current peaks (Supplementary Fig. S3), indicating the enhanced local component fluctuations. The enhanced relaxation behavior exhibiting more diffuse phase transitions and weaker temperature dependence of dielectric constant with increasing BNT can be revealed by temperature-dependent dielectric spectra (Supplementary Fig. S4). All samples show the superparaelectric state with low loss characteristic, where ultrasmall weakly coupled polarization nanoclusters exist[30,31]. Specifically, the temperature coefficient of capacitance ($\Delta C_p/\Delta C_{p,25\,°C}$) can be significantly improved by increasing BNT, enabling the $x = 0.25$ ceramic to satisfy the X9R specification (−55 to 200 °C, $\Delta C_p/\Delta C_{p,25\,°C} \leq \pm 15\%$) for ceramic capacitors while maintaining ultralow dielectric loss ($\leq 0.01$)[32,33]. The above phenomena represent great potential for $x = 0.25$ ceramic to achieve excellent energy storage performance.

### Hierarchical heterostructures

To validate our ideas, we describe the internal structure using transmission electron microscope (TEM) and selected area electron diffraction (SAED). We find striped nanodomains with weak contrast in $x = 0$ ceramic, which is confirmed to be incommensurate modulated antiferroelectric structures exhibiting uniform fringes with alternating widths of 5 and 6 unit cells, as well as satellite spots of (001)/5 and (001)/6 (Supplementary Fig. S5)[34]. Interestingly, two heterostructure regions at the micro-scale, named regions I and II (II$_a$ and II$_b$), can be discovered and separated by SAED patterns in the grains of $x = 0.25$ (Fig. 2a–c and Supplementary Fig. S6). The elongated host lattice diffraction points along [010]$_c$ and [001]$_c$, together with (011)/2 superlattices, can be detected in region I. The region II can be further distinguished as regions II$_a$ and II$_b$ by vertically elongated (001)/2 and (010)/2 superlattices, respectively. Further observation reveals that vertically intersecting striped nanodomains rather than disordered blotched nanodomains (regions II$_a$ and II$_b$) can be clearly found in region I (Fig. 2d). However, blotched nanodomains can also be captured in small amounts in region I, indicating that the coexisting striped and blotched nanodomain heterostructure has been established at the mesoscale (Supplementary Fig. S7). It is well established that blotched nanodomains are an important feature for relaxors[35]. The emergence of vertically intersecting striped nanodomains means the disruption of long-range antiferroelectric ordering, especially in the regions showing disordered blotched domains, which also signifies the enhanced ferroelectirc distortion with increasing BNT.

The designs are more intuitively reflected in the local-scale fluctuations of cation displacements and $BO_6$ tilts. We first characterize the atomic-scale microstructure using atomic-resolution scanning transmission electron microscope (STEM) with annular bright-field (ABF) mode to study $BO_6$ distortions of $x = 0.25$ (Fig. 2e-g). We identify the long-range ordering characteristic with small disturbances of strictly periodically alternating clockwise and anticlockwise $BO_6$ tilts in region I, which can be further proved by the similar clockwise (Average tilt angle: $A_{c\text{-ave}} = 6.43°$) and anticlockwise tilt angles ($A_{a\text{-ave}} = −6.28°$) as well as symmetric distribution behavior from the statistical data (Fig. 2e, g). The ordered $BO_6$ tilt structure is similar to that of the CZ modified NN ceramic exhibiting strong (anti)ferrodistortion

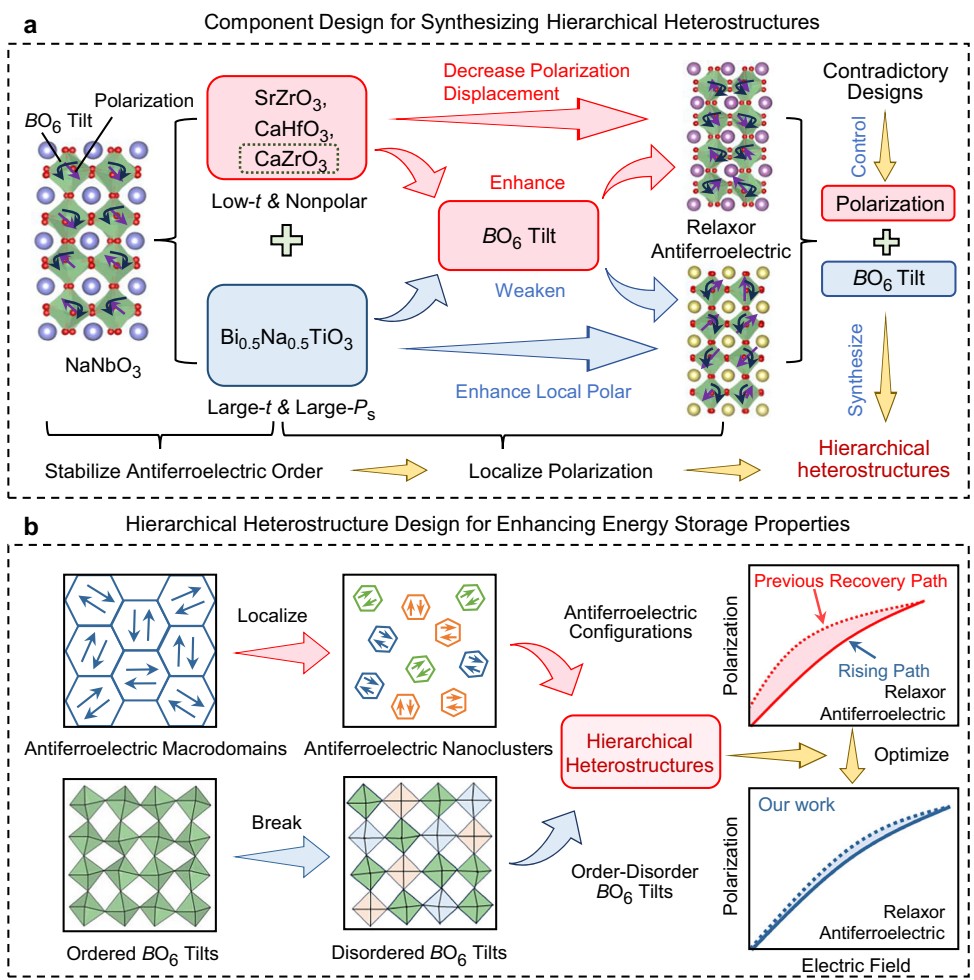

**Fig. 1 | Schematic diagram of designing hierarchical heterostructures for enhancing energy storage properties. a** Component design strategy for designing hierarchical heterostructures[21]. **b** Designing hierarchical heterostructures to optimize polarization evolution paths in NaNbO₃-based lead-free antiferroelectrics.

characteristics[20], which contributes to the emergence of (011)/2 superlattices. Disordered $BO_6$ tilts showing irregular alternations of clockwise and anticlockwise rotation, however, can be detected in region II accompanied by randomly distributed tilt angles with a large difference ($A_{c\text{-ave}}$ = 2.11° and $A_{a\text{-ave}}$ = −1.39°, Fig. 2f, g). Compared to ordered $BO_6$ tilts, the disordered $BO_6$ tilts with largely decreased tilt angle verify the enhanced composition disorder and weakened $BO_6$ distortion. Therefore, we have established a heterodistortion exhibiting order-disorder $BO_6$ tilts at a local scale.

We also explore the local polarization configurations of $x$ = 0.25 by analyzing cation displacements using high-angle annular dark-field (HAADF) STEM (Fig. 3 and Supplementary Fig. S8–9). In region I, we can observe vertically intersecting polarization nanoclusters in different directions, manifested as each antiferroelectric nanoclusters are vertically embedded with polarization regions in other directions, which form polarization interlocking structure (Fig. 3a, e and Supplementary Fig. S8). Combined with the striped nanodomains with different widths (Supplementary Fig. S7), the disturbed antiparallel polarization regions with periodicities of $n$ = 5 to 7 demonstrate the existence of antiferroelectric nature in region I (Supplementary Fig. S8–9), contributing to the elongated host lattice diffraction points that are a series of satellite spots. In region II, we can find alternating striped polarization regions with periodicities of $n$ = 2 along $[001]_c$ and short-range alternating striped polarization regions with multiple periodicities along $[010]_c$. These features display fishbone polarization configuration derived from incommensurate supercells, which

accounts for the appearance of (001)/2 superlattices with vertical elongation characteristic (Fig. 3c, f and Supplementary Fig. S8). The ultrafine fishbone polarization structure, coupled with disordered $BO_6$ tilts, exhibits a relaxation characteristic of blotched domain morphologies at the mesoscale. Note that the antiferroelectric nanoclusters featuring interlocked polarization structure and fishbone polarization configuration are different from traditional nanodomains or PNRs in relaxor ferroelectrics, which typically exhibit ellipsoidal shapes free in nonpolar matrices[35–37]. The distinctive antiferroelectric nanoclusters are expected to enable large polarization fluctuations while delaying polarization saturation. Furthermore, the polarizations in region I are approximately uniformly distributed at various angles, which is related to the interlocking characteristics of the different polarization regions and cannot be found in region II (Fig. 3g). The polarization magnitudes in regions I and II are randomly distributed, revealing the polar regions embedded in the nonpolar matrix (Fig. 3b, d). The approximately equal average polarization magnitudes ($M_{ave}$) and statistical distribution behaviors in regions I and II indicate that the local diverse polarization heterostructure can be formed through roughly similar polarization units arranged and combined in different directions and positions (Fig. 3g, h). To sum up, the hierarchical heterostructures, without obvious element segregation (Supplementary Fig. S10), signifying at the micro-scale (grains), mesoscale (domains), and local scale ($BO_6$ tilts and polarizations), are successfully established in $x$ = 0.25 ceramic by synergistically controlling polarization and $BO_6$ tilt.

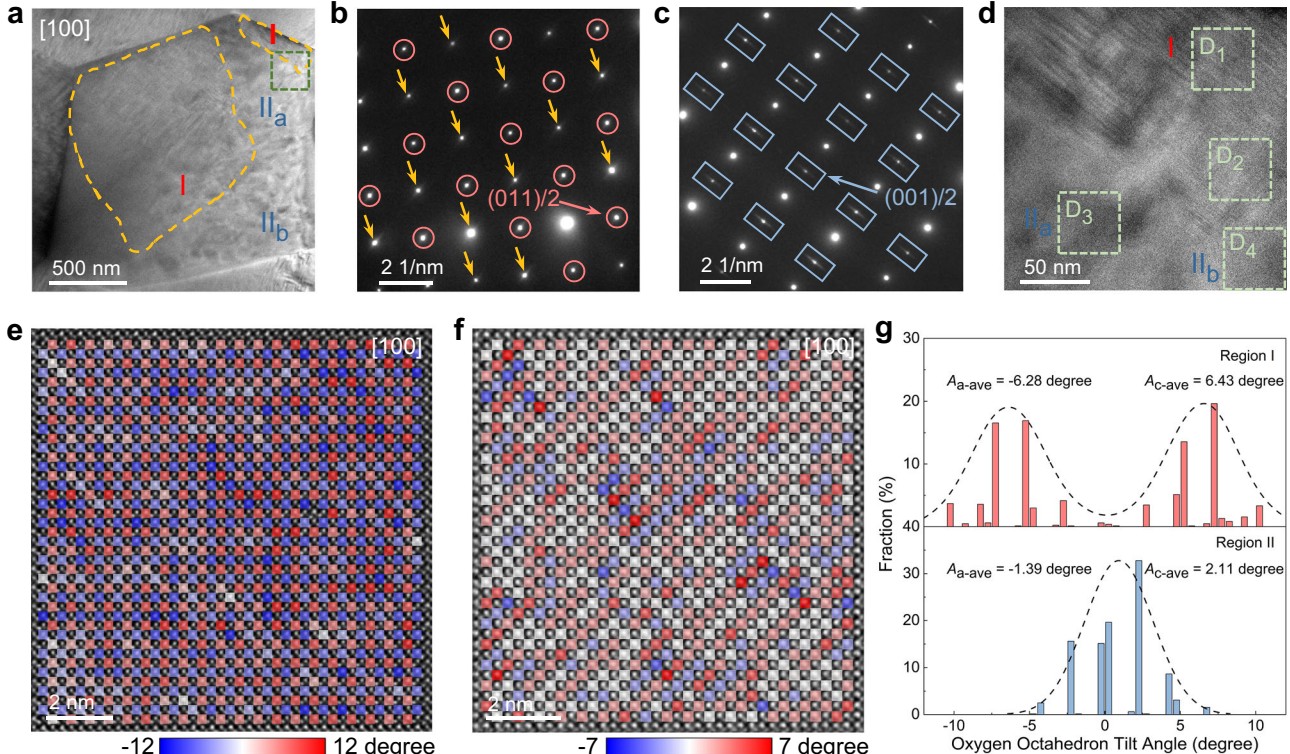

**Fig. 2 | Hierarchical heterostructures and $BO_6$ tilts. a** Heterostructure in one grain for $x = 0.25$ along $[100]_c$. Yellow dashed areas are named region I, while the rest is designated as region II, which is distinguished into two regions, $II_a$ and $II_b$, based on the orientation of the superlattice diffraction points. SAED patterns of regions (**b**) I and (**c**) $II_a$ from (**a**). Yellow arrows, red circles and blue rectangles represent the elongated host lattice diffraction points, (011)/2, and vertically

elongated (001)/2 or (010)/2 superlattice diffraction points, respectively. **d** Amplified domain morphologies of the marked green square area from (**a**). Atomic-resolution ABF-STEM images and the calculated $BO_6$ tilts for regions (**e**) I and (**f**) II. Red, blue, and white square indicate the clockwise, anticlockwise, and no tilt of $BO_6$, respectively. **g** Statistical distributions of $BO_6$ tilt angles for regions I and II.

## Energy storage performance

To explore the full potential of the synthesized NN-CZ-$x$BNT ceramics for applications, we investigate their energy storage properties under electric fields until breakdown from $P$-$E$ loops (Fig. 4a and Supplementary Figs. S11–12). The $U_e$ reaches a maximum of 21.0 J cm$^{-3}$ under 90 kV mm$^{-1}$ for $x = 0.25$ with hierarchical heterostructures exhibiting the desired polarization evolution path, which is approximately four times the energy density ($U_e = 5.6$ J cm$^{-3}$) of $x = 0$ ceramic with long-range incommensurate modulated antiferroelectric structure, especially with the large improvement of $\eta$ from 77% to 90%. In general, dielectrics with large energy density but low efficiency will lead to severe energy dissipation and subsequent thermal effect, significantly shortening the service life of the capacitors[4,12]. Compared to other samples, it can be noted that the $\eta$ of $x = 0.25$ ceramic presents the minimum fluctuation during the continuous increase of electric field and remains above 90% until breakdown. Specifically, in comparison with other bulk ceramic dielectrics including lead-free and lead-based systems, $x = 0.25$ ceramic exhibits significant superiorities for overcoming the bottleneck of simultaneously achieving ultrahigh $U_e$ ($\geq 20$ J cm$^{-3}$) along with ultrahigh $\eta$ ($\geq 90\%$) (Fig. 4b, Supplementary Fig. S13, and Table S1)[7,14,19,38–42], making a breakthrough progress in comprehensive energy storage performance.

## Enhancement mechanisms of capacitive energy storage

The excellent capacitive energy storage should be attributed to the enhanced $E_b$, increased polarization fluctuation, and delayed polarization saturation behaviors with minimal hysteresis. The statistical $E_b$ derived from Weibull distribution is 96 kV mm$^{-1}$ for $x = 0.25$, which is more than twice that (45 kV mm$^{-1}$) for $x = 0$ ceramic (Fig. 4c). The

Weibull modulus $\beta$ values of all ceramics are larger than 10, demonstrating the high sample quality and high reliability. According to the exponential decay relationship between $E_b$ and average grain size $G_a$[43], the decreased $G_a$ from 3.18 ($x = 0$) to 2.05 μm ($x = 0.25$) serves as the vital external contribution to enhance $E_b$ by increasing the density of high-resistance grain boundaries (Supplementary Fig. S14), which can build depletion space charge layers acting as barriers for the migration of charge carriers. Furthermore, thermal breakdown strength can be improved by the less heat generation caused by low dielectric loss, small polarization hysteresis, and high $\eta$. Compared to energy storage ceramics, it is widely recognized that higher $U_e$ can be universally obtained in thin films through achieved higher $E_b$ values[2]. The $x = 0.25$ ceramic exhibits outstanding breakdown resistance strength and a larger $E_b$ than that of the most reported energy storage ceramics[4,15], which can build a bridge to achieve high energy storage properties closer to thin films.

The polarization fluctuation, saturation, and hysteresis behaviors can be effectively controlled by synthesizing hierarchical heterostructures to optimize the polarization evolution pathways. Firstly, the large $P_m$ can be stimulated by the striped nanodomains, while the blotched nanodomains induce small $P_r$[44,45]. In addition, the presence of blotched nanodomains within the striped ones reduces stress during domain switching, leading to the enhanced polarization texture along high electric fields and yielding large $P_m$[4]. Secondly, the polarization interlocking structure exhibits remarkable capability in simultaneously enhancing $P_m$, reducing $P_r$, and delaying polarization saturation with minimal hysteresis. The anomalous increase in polarization with increasing electric field should be attributed to the gradual release of polarization interlock driven by high electric field (Supplementary

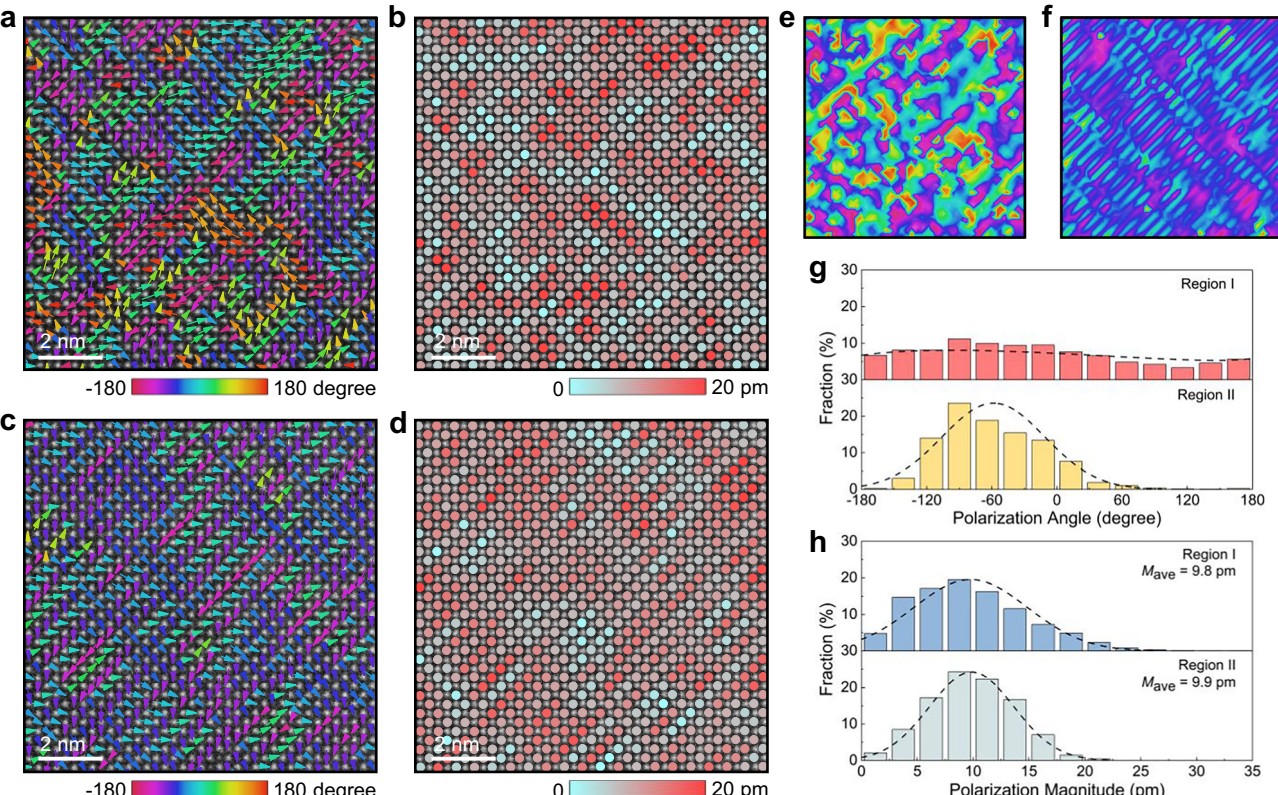

**Fig. 3 | Local polarization structure.** Atomic-resolution HAADF-STEM image of the $x = 0.25$ ceramic with the corresponding (**a**) cation displacement vectors showing polarization interlocking structure and (**b**) polarization magnitudes for region I. Atomic-resolution HAADF-STEM image of the $x = 0.25$ ceramic with the corresponding (**c**) cation displacement vectors showing fishbone polarization configuration and (**d**) polarization magnitudes for region II. Two-dimensional contours of polarization angles for regions (**e**) I and (**f**) II. The color bars are the same as those in (**a**) and (**c**). Statistical distributions of (**g**) polarization angles and (**h**) polarization magnitudes for regions I and II.

Figs. S11 and S15–16), which accelerates more polarization units to align along the direction of electric field and significantly enhances $P_m$, conquering the inverse correlation between the $P_m$ and $E_b$[46]. Moreover, the presence of antiferroelectric nanoclusters with interlocking characteristic delays polarization saturation during charging process. This structure also provides a polarization restoring force after removal of the electric field, facilitating a quick and efficient return to the initial state (Supplementary Fig. S16). The optimized polarization recovery path can largely eliminate the hysteresis loss caused by polarization rotation and the (diffuse) antiferroelectric-ferroelectric phase transition in (relaxor) antiferroelectrics during discharging progress, thereby significantly reducing $P_r$ and improving $\eta$ (Supplementary Figs. S11 and S16–17). Thirdly, fishbone polarization structure in region II can also contribute to high $P_m$ and small $P_r$, but its effectiveness in delaying polarization saturation is inferior to that of polarization interlocking structure. This limitation can be proved by the almost disappeared satellite spots perpendicular to the (010)/2 superlattice points under low electric fields (Supplementary Fig. S16). Lastly, the heterostructure of order-disorder $BO_6$ tilts can absorb partial electric energy and provide excess resistance for forming long-range polarizatin order under electric fields[4,20], delaying polarization saturation. The disordered $BO_6$ distortion with small tilt angles effectively compensates for the deficiency of ordered oxygen distortion with large tilt angles, which can enhance $P_m$ and decrease $P_r$[4,20], thereby achieving a good balance between polarization fluctuation and saturation regulation. Consequently, an optimal polarization evolution path for excellent comprehensive energy storage performance can be achieved in hierarchical-heterostructure relaxor antiferroelectrics.

## Stability and charge-discharge properties

Stability and charge-discharge characteristics are also the important application index for dielectric capacitors. The $x = 0.25$ ceramic shows good stability over a wide temperature range (25 to 100 °C) with $U_e$ variation of <4% and $\eta$ variation of <2% under 60 kV mm$^{-1}$, which is related to the temperature-insensitive dielectric constant and phase structure (Fig. 4d and Supplementary Fig. S18–19). The excellent frequency stability with $U_e$ variation of <7% and $\eta$ variation of <4% can also be obtained at 60 kV mm$^{-1}$ from 1 to 200 Hz (Supplementary Figs. S18 and S20). The increase and decrease in polarization with increasing temperature and frequency, respectively, should be mainly attributed to the polarization interlocking structure, which is easier to unlock at high temperatures and more difficult to unlock at high frequencies (Supplementary Fig. S21). Furthermore, the cyclic fatigue stability of $x = 0.25$ ceramic under a electric field of 60 kV mm$^{-1}$ is also considered (Supplementary Fig. S20). After $10^6$ consecutive electrical cycling tests, the variations of $U_e$ and $\eta$ for the $x = 0.25$ ceramic are less than 2% and 1%, respectively, demonstrating the outstanding fatigue stability. We can also discover the regular underdamped and overdamped discharge waveforms in $x = 0.25$ ceramic under the electric fiels from 10 to 40 kV mm$^{-1}$ (Supplementary Fig. S22). The $P_D$ and discharge energy density $U_D$ increase with increasing electric field, which reach the maximum of 460 MW cm$^{-3}$ and 7.0 J cm$^{-3}$ at 40 kV mm$^{-1}$, respectively (Fig. 4e, f). The ultrahigh $U_D$ can be released by 90% in a very short time of 32 ns, making a good application prospect for advanced capacitors (Supplementary Table S2).

In summary, we develop hierarchical heterostructures in NaNbO$_3$-based lead-free antiferroelectrics by synergistically controlling cation

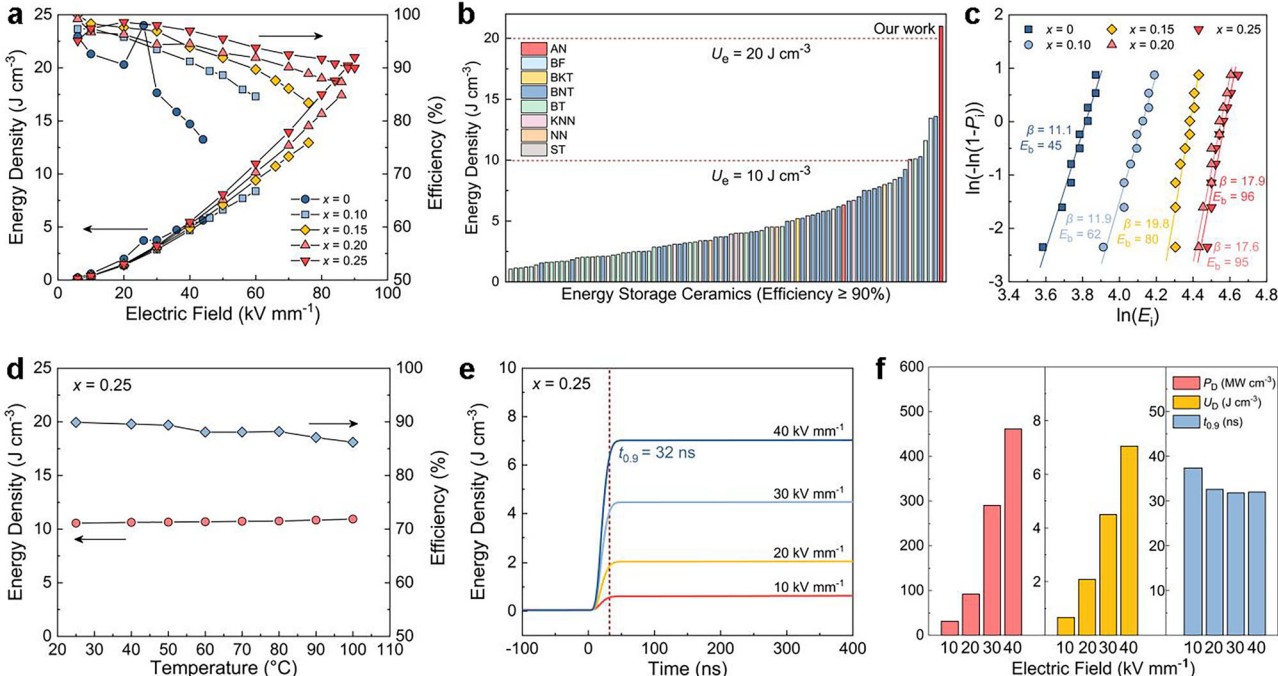

**Fig. 4 | Energy storage and charge-discharge performance of the NN-CZ-xBNT ceramics. a** Energy density and efficiency of the NN-CZ-xBNT ceramics. **b** Comparisons of energy density between our ceramic and other reported lead-free bulk ceramics with $\eta \geq 90\%$ and $U_e \geq 1.0$ J cm$^{-3}$ (AN: AgNbO$_3$, BF: BiFeO$_3$, BKT: Bi$_{0.5}$K$_{0.5}$TiO$_3$, BT: BaTiO$_3$, KNN: (K,Na)NbO$_3$, ST: SrTiO$_3$). The references are provided in Supplementary Table S1. **c** Weibull distribution analysis of the breakdown fields of the NN-CZ-xBNT ceramics. **d** Temperature-dependent energy storage performance of $x = 0.25$ at an electric field of 60 kV mm$^{-1}$. **e** Time-dependent overdamped discharge energy density of $x = 0.25$. **f** Charge-discharge performance under different electric fields of $x = 0.25$.

displacements and $BO_6$ tilts to form antiferroelectric nanoclusters with polarization interlocking structure and fishbone polarization configuration, as well as order-disorder $BO_6$ tilts at local scale. The hierarchical heterostructures can effectively optimize polarization evolution pathways to increase polarization fluctuation and delay polarization saturation with minimal hysteresis, achieving a large improvement of overall energy storage performance in ceramic capacitors. The strategy should be applicable to other material systems such as thin films and composites. Generally, this strategy can be extended to other functionalities involving the development of materials through multi-scale heterostructural regulation.

# Methods

## Sample synthesis

A series of $(1-x)(0.9\text{NaNbO}_3\text{-}0.1\text{CaZrO}_3)\text{-}x\text{Bi}_{0.5}\text{Na}_{0.5}\text{TiO}_3$ (abbreviated as NN-CZ-xBNT; $x = 0$ to 0.25) ceramics were synthesized by a conventional solid-state reaction method. High-purity Na$_2$CO$_3$ (Aladdin, 99.99%), Bi$_2$O$_3$ (Aladdin, 99.99%), CaCO$_3$ (Aladdin, 99.9%), Nb$_2$O$_5$ (Aladdin, 99.9%), ZrO$_2$ (Aladdin, 99.99%), and TiO$_2$ (Aladdin, 99.8%) were used as the raw materials, and 0.5 mol% MnO$_2$ (Aladdin, 99.0%) was used as a sintering aid. The dried raw materials were mixed at the designed stoichiometric ratios by planetary ball milling with alcohol for 24 h and then calcined twice at 800 °C for 5 h. Subsequently, the calcined powders were remixed with 0.5 wt% PVB binders through high-energy ball milling under 700 r min$^{-1}$ for 15 h. The mixed powders were uniaxially pressed into pellets with diameters of 10 mm under 400 MPa. After being covered with sacrificial powders of the same compositions, the pellets were sintered at 1200–1390 °C in closed double crucibles for 2 h at a heating rate of 5 °C min$^{-1}$ after excluding PVB binders at 550 °C for 2 h. The surface of ceramics was carefully polished and coated with silver electrodes, which were fired at 550 °C for 30 min for measuring electrical properties.

## Structure characterizations

The phase structures of the ceramic powders were detected using an X-ray diffractometer (XRD, Smartlab, Rigaku) with Co target ($\lambda = 1.79$ Å). The room-temperature neutron total scattering data were collected at Nanoscale-Ordered Materials Diffractometer (NOMAD) in the Spallation Neutron Source (SNS), Oak Ridge National Laboratory. The grain morphologies were detected through a scanning electron microscope (SEM, LEO1530, ZEISS SUPRA 55), and grain size was analyzed by Nano Measure software. The finely polished ceramics below 40 μm were thinned using an ion milling system (PIPS, Model 691, Gatan Inc., Pleasanton) with a liquid nitrogen-cooled stage for transmission electron microscope measurement. Domain morphologies, selected area electron diffraction (SAED), and element distribution analysis were performed on a field-emission transmission electron microscope (TEM, JEM-2100, JEOL, Japan) with an accelerating voltage of 200 kV. The atomic-scale HAADF-STEM and ABF-STEM images were carried out on Cs-corrected Hitachi HF5000 microscope. The microscope settings were: probe size in UHR mode and convergence semi-angle of 20 mrad, and collection semi-angle of 60–320 mrad (HAADF) and 11–22 mrad (ABF). The images were acquired under conditions of fast scanning and cross-correlation summing of multiple frames to minimize sample drift. The atomic column positions at picometer-precision fitting were performed using the MATLAB code for calculating the polarization vectors with different polarization magnitudes and polarization angles, as well as oxygen octahedral tilts.

## Electrical property characterizations

The room-temperature, temperature- and frequency-dependent $P$-$E$ loops as well as room-temperature $J$-$E$ loops of the sintered ceramics with a thickness of about 35-60 μm and an electrode diameter of about 0.8-1.0 mm were tested using a ferroelectric analyzer (aix ACCT, TF Analyzer 1000). Temperature-dependent capacitance and dielectric

spectra were measured using a precision LCR meter (Keysight E4990A). The charge-discharge performance under different electric fields was characterized by a commercial charge-discharge platform (CFD-003, Gogo Instruments Technology).

## Weibull distribution

The Weibull experiments of the NN-CZ-$x$BNT ceramics can be calculated by the following equations:

$$P_i = i/(n+1) \tag{1}$$

$$X_i = \ln(E_i) \tag{2}$$

$$Y_i = \ln(\ln(1/(1-P_i))) \tag{3}$$

where $E_i$ is the specific breakdown electric field of each sample, $i$ means the ordinal number of each sample, and $n$ is the total amount of ceramic for each sample ($n = 10$ in this work). The intersection of the fitted line and $Y_i = 0$ should be the theoretical $E_b$ value. Weibull parameter $\beta$ evaluates the distribution of $E_i$.

## Data availability

All data supporting this study and its findings are available within the article and its Supplementary Information. Any data deemed relevant is available from the corresponding author upon request.

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

## Acknowledgements

This research used resources at the Spallation Neutron Source, a DOE Office of Science User Facility operated by the Oak Ridge National Laboratory. This work was supported by the National Key Research and Development Program of China (grant no. 2023YFB3508200), the Beijing Outstanding Young Scientist Program (grant no. JWZQ20240101015), the National Natural Science Foundation of China (grant nos. 22235002 and 22575018), the China National Postdoctoral Program for Innovative Talents (grant no. BX20230040), and the China Postdoctoral Science Foundation (grant no. 2023M740208).

## Author contributions

This work was conceived and designed by L.C., H.Q., and J.C. L.C. fabricated the samples, tested the energy storage, dielectric, structure, stability, and other properties, and processed related data, assisted by H.Q. The XRD and dielectric spectra were collected by H.Y. The SEM, TEM, and STEM images were filmed and processed by L.C., T.H., and Z.F. The manuscript was drafted by L.C. and revised by H.Q., S.Z., and J.C. All authors participated in the data analysis and discussions.

## Competing interests

The authors declare no competing interests.
