## [Transparent Peer Review file · Nature Communications]

Design of hierarchical-heterostructure antiferroelectrics for ultrahigh capacitive energy storage

Corresponding Author: Professor Jun Chen

Version 0:

Reviewer comments:

Reviewer #1

(Remarks to the Author)

This is an interesting work reporting high energy storage density and efficiency enabled by utilizing relaxor antiferroelectrics. The authors suggested a strategy of combining polarization resulting from ionic displacements and oxygen octahedra tilting to achieve high energy storage density as well as high efficiency. They conducted comprehensive STEM analysis to understand the structural mechanism. The reviewer believes that this work can be reconsidered after a major review. However, the authors MUST clarify their technological advances distinguished from their own and other's previous works. Detailed comments can be read below.

-The two corresponding authors reported high energy storage using NN-BNT relaxor antiferroelectrics in Nature Communications in January, 2025.[<https://doi.org/10.1038/s41467-025-56316-9>] What are the fundamental differences distinguished from the aforementioned authors' previous work. For the reviewer, except the differences in materials (Now the authors added CsZrO₃ to NN-BNT.), the fundamental idea seems quite similar at first glance. Please clearly explain the fundamental differences in the main idea and results.

-In the previous work mentioned in the former comment, the authors explained the phenomenon with the core-shell structure. However, now the argued mechanism seems different from the core-shell structure. Why adding a few % CsZrO₃ change the mechanism of the energy density enhancements?

-There was a report on an antiferroelectric/relaxor composite for energy storage applications. [J. Mater. Chem. C, 2020, DOI: 10.1039/D0TC00589D] Can the authors compare their work to the previous work? For high impact journals such as Nature Communications, significant technological progress is essential. It should be also described in the main text.

-Many readers would not be familiar with the term "relaxor antiferroelectric". This concept should be compared to the relaxor ferroelectric with adequate schematic figures at least in SI. Moreover, the reviewer could not be sure whether using the term "polar nano regions" for relaxor antiferroelectrics. From the reviewer's postulation there may be "antipolar nano regions" as zero electric field and they would be poled and grow under sufficiently high external field.

-Generally, relaxor ferroelectric is strongly temperature and frequency dependent. A strong temperature dependent changes in dielectric constant is one typical characteristics of relaxor ferroelectric. The reviewer would expect similar nature of relaxor antiferroelectric. Please explain how the weak temperature dependence can be observed for the relaxor antiferroelectric in this work.

-In figure 4a, the authors report maximum energy density higher than 20 J/cm³ with applying ~90 kV/mm. However, they examined the temperature stability, discharge energy density at a field of ~40 to ~60 kV/mm. The reviewer does not believe that it is fair to present the energy density which can be reliably utilized. The authors cannot agree that the benchmark graph in figure S14 is fair.

-Provide endurance test results to prove reliable energy storage applications.

Reviewer #2

(Remarks to the Author)

This manuscript presents a major advance in dielectric energy storage by designing a hierarchical-heterostructure NaNbO_3 -based antiferroelectric ceramic, achieving an ultrahigh recoverable energy density of 21.0 J cm^{-3} at 90% efficiency. The authors leverage multi-scale structural control—including interlocked and fishbone nanodomains and BO_6 tilt modulation—to delay polarization saturation and suppress hysteresis, outperforming existing lead-free bulk ceramics.

- A lead-free bulk ceramic achieving record-high U_e (21.0 J/cm^3) and 90% η , alongside ultrafast discharge (32 ns) and high-power density (460 MW/cm^3), marking a significant improvement over prior systems ($\sim 12 \text{ J/cm}^3$ max for NaNbO_3 -based ceramics).
- Yes, this work is highly significant; it clearly surpasses previous benchmarks (e.g., Qi et al., *Adv. Funct. Mater.*, 2019) in both energy density and efficiency for bulk lead-free ceramics. It introduces a broadly applicable design strategy with potential relevance to other energy storage platforms.
- The manuscript's experimental data robustly supports its conclusions. However, additional evidence such as fatigue endurance, batch reproducibility, and comparative benchmarking would further reinforce key claims.
- No fatal flaws, but gaps remain in scalability discussion, long-term reliability, and modeling of mechanisms. These should be addressed for higher-impact publication but do not prohibit publication.
- Yes, the experimental approach—including TEM, P–E loops, dielectric and breakdown testing—is sound and meets the standards of the field.
- While general processing details are provided, more explicit data on sample-to-sample consistency, synthesis repeatability, and sintering control is needed to confirm reproducibility.

Comments and Scope for Improvement

1. Add a summary table comparing U_e , η , E_b , and $t_{0.9}$ with prior top-performing lead-free systems.
2. Explicitly quantify improvement over established NaNbO_3 -based ceramics and thin-film benchmarks.
3. Add charge-discharge cycling data (e.g., 10^4 – 10^6 cycles).
4. Support the proposed polarization unlocking and restoring force mechanisms with basic phase-field simulations or schematic energy landscapes.
5. In discussion, acknowledge that thin films (e.g., Pan et al., *Science*, 2019) can achieve higher U_e via higher E_b . Position the work as leading within bulk ceramics, with paths to bridge toward film-like performance.
6. Include permittivity vs. E curves in supplementary.
7. Simplify overly long sentences. Add subheadings in the Results section for readability. Define terms like “interlocking” and “fishbone polarization” when first used.

Reviewer #3

(Remarks to the Author)

The authors demonstrate the feasibility of enhancing energy storage performance by designing the composition NN-CZ-xBNT, synthesized via a conventional solid-state route. The topic of the manuscript is interesting and timely. However, in the current format, the manuscript faces a lot of issues such as structural imbalance and in-depth explanation. This is mandatory for a reputed journal like *Nat. comm.* Therefore, the manuscript requires a major revision, and the following points must be addressed before considering for publication.

Comments

1. As we are moving towards the miniaturized electronics world, the authors are expected give the importance of ceramics capacitors over the thin film capacitors in application point of view in the introduction.
2. The authors mentioned “Thus, tailoring the polarization behavior of RFE ceramics is crucial to achieving high energy storage performance (ESP) and meeting the growing demands of advanced capacitor technologies”. It is anticipated to give why RFE ceramics are needed for advanced capacitor technology.
3. Still, it is unclear in the introduction section that how BNKT-NN28 ceramic capacitors compete with MLCCs and thin films. Please go through the following work [<https://doi.org/10.1016/j.jmat.2024.100980>]. Also, I suggest the authors to make a comparison table of U_e and U_f with other BNT ceramic based systems in the final part of result and discussion.
4. I suggest the authors to follow the section division as per journal guideline. For instance, it is given subheading likes “General structure characterizations” and “Hierarchical heterostructure and BO_6 tilts”, which feels inappropriate. So, please check it and make it more simple headings.
5. The authors proposed a hierarchical heterostructure. However, the work NN-CZ-xBNT suggest doping. The authors need to give a proper explanation why they chose the term heterostructure. Also, in Fig. 1, the mechanism of forming nanoclusters featuring, the so-called interlocked polarization structure and fishbone polarization configuration is unclear. For instance, it is unclear how the ordered BO_6 tilts forms nanocluster. In the supplementary Fig. S2 also, it is not clear. The authors need to say how CZ and BNT transforms the macrodomains to these mentioned nanoclusters. Instead of simply providing low-t and non-polar. Otherwise, the authors meant PNR, it is agreeable. If it is not, the authors need to give literature or theoretical evidence to prove these terms, especially, fishbone polarization configuration. This is the core part of the original article and should be very clear to the audience. Therefore Fig. 1 must be reader friendly with precise explanation. This is lacking here.
6. However, the work ref [30] cited by the reviewers calculated the γ value for confirming the relaxor nature. I suggest the authors to do the same.
7. The authors stated that “All samples show the superparaelectric state with low loss characteristic, where ultrasmall weakly coupled PNRs exist.” It should be better to correlate how these ultrasmall weakly coupled PNRs enhance the diffused phase

transition and relaxation behaviour. Because by seeing the temperature dependent dielectric spectroscopy graph, it is visible that the room temperature permittivity is significantly reduced (almost 2 times). And the P-E loops in the Fig. S12 does not follow this trend. At the low electric fields, also the PNRs nearly maintained almost same saturation polarization compared to $x=0$ composition, indicating a good response of PNRs to electric field at room temperature. Therefore, I suggest the authors to give a proper explanation for this.

8. The authors used the term “fishbone polarization configuration” multiple times. However, I could not find such terms in the cited literatures [34-36]. If the authors are the first to introduce the term, it should be well defined. For example, the authors should clearly define how this fishbone polarization configuration is different from nanodomains like “polymorphic nanodomains”. Because whatever the authors mentioned in the literature are features pointing towards either PNRs or polymorphic nanodomains.

9. Finally, the structure of the manuscript should be well-defined and should be written reader friendly. Please follow the journal guidelines.

Version 1:

Reviewer comments:

Reviewer #1

(Remarks to the Author)

The manuscript is ready for publication.

Reviewer #2

(Remarks to the Author)

The authors have incorporated my suggestions and given an explanation/rebuttal wherever required. I recommend the manuscript be considered for publication.

Reviewer #3

(Remarks to the Author)

I appreciate the efforts of the authors. But the authors were unable to answer the comments properly and therefore, I reject the paper. For instance, to my comment 1, the authors mentioned “From a practical application standpoint, compared with thin film capacitors, ceramic capacitors offer distinct advantages, including simple fabrication, low cost, high power and energy storage capacity, thus attracting growing attention in the field of dielectric energy storage.” This cannot be agreeable as scientific community is giving importance to thin films for the integration and even trying to develop flexible thin films [<https://doi.org/10.1016/j.pmatsci.2022.101046>,<https://doi.org/10.1016/j.jeurceramsoc.2024.02.016>]. However, the high breakdown strength is a big issue with the thin film. On the other hand, ceramic capacitors produce better ESD at low voltages. This will minimize the cost of insulation technology and miniaturization of the electronic devices. I clearly gave the hint to the authors, and they were unable to implement it [<https://doi.org/10.1016/j.jmat.2024.100980>]. Also, I suggested the authors to make a comparison table of U_e and U_f with other BNT ceramic based systems in the final part of result and discussion. This is very important as these Figure of parameters determines how efficient is your ceramics. Even I suggested the authors to make a comparison with BNT based systems and the authors are commenting it is inadequate. Also, the authors selected only a very few works only. Thus, I find the authors response very weak and a renowned journal like Nat. Comm. require high quality analysis and unique results to publish.

Version 2:

Reviewer comments:

Reviewer #2

(Remarks to the Author)

I recommend considering the manuscript for publication.

Dear reviewers:

We highly appreciate the valuable suggestions and comments on our manuscript entitled “**Design of hierarchical-heterostructure antiferroelectrics for ultrahigh capacitive energy storage**” (NCOMMS-25-51936-T). The careful revisions have been made accordingly, which were highlighted in red in the revised manuscript. We hope that these changes adequately address the concerns raised. The responses to the comments are listed point-by-point below:

Reviewer #1 (Remarks to the Author):

This is an interesting work reporting high energy storage density and efficiency enabled by utilizing relaxor antiferroelectrics. The authors suggested a strategy of combining polarization resulting from ionic displacements and oxygen octahedra tilting to achieve high energy storage density as well as high efficiency. They conducted comprehensive STEM analysis to understand the structural mechanism. The reviewer believes that this work can be reconsidered after a major review. However, the authors MUST clarify their technological advances distinguished from their own and other’s previous works. Detailed comments can be read below.

Reply: We highly appreciate the positive and professional comments for the present study.

1. The two corresponding authors reported high energy storage using NN-BNT relaxor antiferroelectrics in Nature Communications in January, 2025. [<https://doi.org/10.1038/s41467-025-56316-9>] What are the fundamental differences distinguished from the aforementioned authors’ previous work. For the reviewer, except the differences in materials (Now the authors added CsZrO₃ to NN-BNT.), the fundamental idea seems quite similar at first glance. Please clearly explain the fundamental differences in the main idea and results.

Reply: Thanks for your good comments. The binary system comprising NN and BNT can form relaxor antiferroelectrics over a wide range, yet they suffer from low energy storage density U_e and efficiency η (*Acta Mater.* **208**, 116710 (2021); *Adv. Mater.* **34**, 2205787 (2022)). In our previous work (*Nat. Commun.* **16**, 886 (2025)), we proposed the strategy of polymorphic heterogeneous shell utilizing NN with a high sintering temperature and BNT with a low sintering temperature to efficiently fabricate core-shell structures through optimized preparation (including ball milling and sintering) processes, thereby enhancing breakdown behavior. Building on this, we leveraged the diffusion effects of various elements

to tailor highly polarized cores and polymorphic polarized shells, achieving, for the first time, local polarization structure control in dual-phase dielectrics. This approach enables the synergistic regulation of maximum and remnant polarization under ultrahigh electric field loading and unloading conditions, respectively. Consequently, we attained high U_e (12.7 J cm^{-3}) and η (87.2%) in 0.5NN-0.5BNT core-shell energy storage ceramics, which is larger than that ($U_e \sim 4.7 \text{ J cm}^{-3}$ and $\eta \sim 85.0\%$) of 0.5NN-0.5BNT ceramic without core-shell structure. However, despite the notable overall performance enhancement, the persistent issue of low η in (relaxor) antiferroelectric remaining below 90% for high U_e has not been resolved, leading to substantial energy loss.

In this work, we employ non-polar CZ with a low tolerance factor to stabilize the antiferroelectric characteristics, simultaneously disrupting long-range ferroelectric order while regulating oxygen octahedral tilt (enhancing effect)—a feature not explored in our previous work (*Nat. Commun.* **16**, 886 (2025)). Based on this foundation, we incorporate high-spontaneous polarization BNT with a large tolerance factor (forming a 0.75(0.9NN-0.1CZ)-0.25BNT solid solution) to further enhance local polarization and mitigate oxygen octahedral tilt. By integrating two components with distinct properties to tailor the local structure of antiferroelectrics and designing hierarchical heterostructures, we have achieved a groundbreaking advancement in energy storage performance ($U_e \sim 21.0 \text{ J cm}^{-3}$ and $\eta \sim 90\%$). Notably, we have, for the first time, systematically unveiled the local polarization configuration with antiferroelectric features in relaxor antiferroelectrics, achieving structural heterogeneity at the local scale and elucidating a novel mechanism for enhancing energy storage performance. Consequently, the two studies exhibit marked differences in conceptual design, innovation, structural composition, and performance outcomes.

2. In the previous work mentioned in the former comment, the authors explained the phenomenon with the core-shell structure. However, now the argued mechanism seems different from the core-shell structure. Why adding a few % CaZrO_3 change the mechanism of the energy density enhancements?

Reply: Thanks for your good comments. Firstly, there is a significant difference in the composition between the two works, with 0.5NN-0.5BNT and 0.75(0.9NN-0.1CZ)-0.25BNT. Not only is there a small amount of CZ, but the BNT content is also significantly different. Secondly, the preparation processes are also significant different. The former utilizes the differences in sintering characteristics to separately prepare NN and BNT. After calcination and secondary ball milling, they are mixed by high-

energy ball milling for a short period of 20 minutes to form 0.5NN-0.5BNT solid solutions with a core-shell structure. The latter adopts the general solid-phase method for preparation, which involves mixing all the raw materials in the first ball milling process. The significant differences in preparation processes are also an important reason for the differences in multi-scale structures. In addition, the solid solution limit of CZ in NN based ceramics is around 10%, and excessive introduction can also form second phases leading to new structural differences. Thirdly, the continuous introduction of BNT significantly weakens the oxygen octahedral tilt during the enhanced relaxation process and does not bring about a strengthening effect. As the regulation of oxygen octahedral tilt has a significant impact on energy storage performance, the introduction of ultralow-tolerance factor CZ takes into account the strengthening of oxygen octahedral tilt during the relaxation process, resulting in the coexistence of ordered and disordered oxygen octahedral tilt configurations. Fourthly, the introduction of high-polarization BNT and non-polar CZ can also cause differences in local polarization control, resulting in different local antiferroelectric polarization configurations. Finally, the introduction of CZ with ultralow tolerance factor can maintain the antiferroelectric characteristics (diffusion phase transition and large P_m) as much as possible during the relaxation process, ensuring the formation of relaxor antiferroelectrics and providing a guarantee for unlocking the polarization configuration of relaxor antiferroelectrics to enhance energy storage performance. Therefore, the introduction of a small amount of CZ combined with component design and preparation processes can form a new type of local structures and alter the mechanism of energy storage enhancement.

3. There was a report on an antiferroelectric/relaxor composite for energy storage applications. [J. Mater. Chem. C, 2020, DOI: 10.1039/D0TC00589D] Can the authors compare their work to the previous work? For high impact journals such as Nature Communications, significant technological progress is essential. It should be also described in the main text.

Reply: Thanks for your good suggestions. The previous work has established novel relaxor/antiferroelectric composites with multi-phase structure of an antiferroelectric embedded in a relaxor ferroelectric, which can contribute to high energy storage performance ($U_e \sim 2.85 \text{ J cm}^{-3}$ and $\eta \sim 80\%$) (*J. Mater. Chem. C*, **8**, 5681-5691 (2020)). Firstly, compared to this previous work, the relaxor antiferroelectric we reported achieved a significant breakthrough in performance ($U_e \sim 21.0 \text{ J cm}^{-3}$ and $\eta \sim 90\%$), with energy storage density increased by over 7 times at efficiencies above 90%. Secondly, the

progressiveness of the design idea and hierarchical heterostructures have been elaborated in detail in the full text. Thirdly, there are many excellent dielectric energy storage works and reports in relaxor antiferroelectrics. In relevant reports, the relaxor antiferroelectric ceramic with hierarchical heterostructures in our work exhibited the optimal polarization evolution path and comprehensive energy storage performance (Fig. R1). Based on your suggestion, in order to further highlight the technological progress, we have compared our work with this previous work and other reports (including lead-free and lead-based systems exhibiting relaxor ferroelectrics and antiferroelectrics) in the manuscript and demonstrated our advantages. The relevant literatures have been cited in the manuscript. The relevant modifications have been added to the manuscript **in red** (Page 5-6, Page S15, Supplementary Fig. S13).

Figure Redacted

Fig. R1

P-E loops and energy storage properties for NaNbO₃-based relaxor antiferroelectrics. *P-E loops for (a)* 0.76NaNbO₃-0.24(Bi_{0.5}Na_{0.5})TiO₃ (*Adv. Funct. Mater.* **29**, 1903877 (2019)), **(b)** 0.90NaNbO₃-0.10BiFeO₃ (*Energy Storage Mater.* **43**, 383-390 (2021)), **(c)** (Na_{0.91}La_{0.09})(Nb_{0.82}Ti_{0.18})O₃ (*ACS Appl. Mater. Interfaces* **12**, 32871 (2020)), **(d)** 0.76NaNbO₃-0.24Sr_{0.7}Bi_{0.2}TiO₃ via Spark Plasma Sintering (SPS) (*Scripta Mater.* **210**, 114428 (2022)), and **(e)** NN-CZ-0.25BNT ceramic in our work. **(f)** A comparison of energy storage performance between NN-CZ-0.25BNT ceramic in our work and other reported relaxor antiferroelectric ceramics.

4. Many readers would not be familiar with the term “relaxor antiferroelectric”. This concept should be compared to the relaxor ferroelectric with adequate schematic figures at least in SI. Moreover, the reviewer could not be sure whether using the term “polar nano regions” for relaxor antiferroelectrics. From the reviewer’s postulation there may be “antipolar nano regions” as zero electric field and they would be poled and grow under sufficiently high external field.

Reply: Thanks for your good comments. Indeed, compared to relaxor ferroelectrics, relaxor antiferroelectrics are not common and familiar to readers. However, The *P-E* loops of relaxor antiferroelectrics and relaxor ferroelectrics exhibit similar shapes with minimal differences, except that the *P-E* loops of relaxor antiferroelectrics generally feature delayed polarization saturation and a bulging polarization recovery path (*Adv. Funct. Mater.* **29**, 1903877 (2019); *Energy Storage Mater.* **43**, 383-390

(2021); *Adv. Mater.* **34**, 2204356 (2022)). It should be noted that these characteristics cannot serve as a basis for identifying relaxor antiferroelectrics. The relaxation behavior of relaxor antiferroelectrics can be determined through dielectric temperature spectra, while their antiferroelectric nature can be verified by observing the superlattice structure via XRD or SAED. Importantly, atomic-resolution STEM can resolve the local structural features of relaxor antiferroelectrics, providing an intuitive revelation of their unique characteristics. Consequently, it is challenging to provide a clear schematic diagram of the P - E loops for relaxor antiferroelectrics to distinguish them from relaxor ferroelectrics. Nevertheless, we have tentatively presented the internal structure of relaxor antiferroelectrics and the optimized P - E loop of relaxor antiferroelectrics in Fig. 1.

In our manuscript, we did not refer to the local polarization structure of relaxor antiferroelectrics as “polar nanoregions”. As you rightly pointed out, such terminology is inappropriate. Instead, the antipolar polarization structure in relaxor antiferroelectrics is significantly disrupted into nanoscale, as evidenced by disturbances in the polarization angle, magnitude, and commensurate structure, while only retaining the local antiferroelectric polarization characteristics. To differentiate it from the antipolar structure of conventional antiferroelectrics, we term it as “antiferroelectric nanoclusters”, which balances the local-scale antiferroelectric polarization characteristics with the nanoscale properties of relaxors. Thanks again.

5. Generally, relaxor ferroelectric is strongly temperature and frequency dependent. A strong temperature dependent changes in dielectric constant is one typical characteristics of relaxor ferroelectric. The reviewer would expect similar nature of relaxor antiferroelectric. Please explain how the weak temperature dependence can be observed for the relaxor antiferroelectric in this work.

Reply: Thanks for your good comments. Generally, the dielectric constant of normal ferroelectrics and antiferroelectrics demonstrate pronounced temperature dependence, manifesting as distinct dielectric anomaly peaks that emerge with temperature variations, which can be attributed to the significant phase transitions. However, relaxor ferroelectrics or relaxor antiferroelectrics exhibit diffuse phase transition characteristics, characterized by relatively flat dielectric constant curves and diminished temperature dependence. The higher the degree of relaxation, the flatter the dielectric curve and the weaker the temperature dependence. This occurs because different ions occupy identical crystallographic sites, enhancing the random field and disrupting the long-range ordered structure, which is replaced by

nanoscale domains or polar regions. According to relaxor theory, distinct nanoregions possess varying Curie temperatures or phase transition temperatures (*J. Mater. Sci.* **41**, 31-52 (2006)). Nanoregions with low phase transition temperatures will preferentially undergo phase transition with increasing temperature. Unlike large-scale ferroelectric/antiferroelectric domains, which undergo significant dielectric constant changes at fixed temperatures (exhibiting strong temperature dependence), these local antiferroelectric nanoclusters can undergo phase transitions at their respective temperature points as the temperature rises. This results in macroscopic manifestations of diffuse phase transitions (with weak temperature dependence) and flat dielectric curves in relaxor antiferroelectrics, which is similar to relaxor ferroelectrics. Additionally, we also supplement temperature-dependent dielectric spectra of ceramic samples measured at various frequencies (Fig. R2). The frequency-independent dielectric constant of all samples can be clearly found at room temperature. The relevant revisions have been added to the Supplementary Information in red (Page S5, Supplementary Fig. S4).

Fig. R2

Dielectric and capacitance properties of the NN-CZ-xBNT ceramics. Temperature- and frequency-

dependent dielectric spectra for (a) $x = 0$, (b) $x = 0.10$, (c) $x = 0.15$, (d) $x = 0.20$, and (e) $x = 0.25$ ceramics. (f) Capacitance properties ($\Delta C_p/C_{p,25^\circ\text{C}}$) of the NN-CZ- x BNT ceramics at 1 kHz.

6. In figure 4a, the authors report maximum energy density higher than 20 J/cm^3 with applying $\sim 90 \text{ kV/mm}$. However, they examined the temperature stability, discharge energy density at a field of ~ 40 to $\sim 60 \text{ kV/mm}$. The reviewer does not believe that it is fair to present the energy density which can be reliably utilized. The authors cannot agree that the benchmark graph in figure S14 is fair.

Reply: Thanks for your good comments. I fully appreciate the reviewer's meticulous attention to the energy storage performance we reported. However, our performance comparison is conducted in a fair and impartial manner, primarily based on the following justifications. Firstly, as depicted in Fig. S13 (previous Fig. S14), the optimal energy storage performance we compared was assessed using the same quasi-static testing principles and reliable universal equipment, with similar testing conditions and sample specifications. The energy storage performance of the NN-CZ-0.25BNT ceramic we reported also demonstrates high repeatability (Fig. R3). Secondly, in the field of dielectric energy storage, the temperature stability or reliable applicability of samples is typically evaluated within the range of $1/2E_b$ to $2/3E_b$ (*Science* **384**, 185-189 (2024); *Science* **365**, 578-582 (2019); *Nat. Commun.* **16**, 807 (2025); *Energy Environ. Sci.* **16**, 4511-4521 (2023)). The dielectric breakdown strength is influenced by factors such as temperature, humidity, and testing frequency. Clearly, achieving E_b at room temperature under varying temperature conditions is practically unattainable. Our test electric field, set at approximately $2/3E_b$, represents a high level of stability testing. Moreover, the temperature stability of the energy storage performance (10.75 J cm^{-3} at 60 kV mm^{-1}) we reported is also in the first order compared to other reported advanced energy storage ceramics (8.64 J cm^{-3} at 50 kV cm^{-1} in *Nat. Commun.* **16**, 807 (2025); 8.5 J cm^{-3} at 50 kV cm^{-1} in *J. Am. Chem. Soc.* **146**, 460-467 (2024); 4.26 J cm^{-3} at 40 kV cm^{-1} in *Acta Mater.* **278**, 120278 (2024); 6.35 J cm^{-3} at 42 kV cm^{-1} in *Energy Environ. Sci.* **16**, 4511-4521 (2023)), demonstrating significant advancement. Thirdly, the discharge energy density (measured through charge-discharge tests) is determined using dynamic testing methods, which are more susceptible to breakdown than quasi-static testing. It is also common for the test electric field to be lower than the E_b measured at room temperature. Furthermore, our charge-discharge performance (7.0 J cm^{-3} at 40 kV mm^{-1}) exhibits considerable superiority over many high-performance ceramics (2.58 J cm^{-3} at 22 kV cm^{-1} in *Nat. Commun.* **16**, 807 (2025); 5.0 J cm^{-3} at 35 kV cm^{-1} in *J. Am. Chem. Soc.* **146**, 460-467 (2024);

4.47 J cm⁻³ at 36 kV cm⁻¹ in *Acta Mater.* **278**, 120278 (2024); 6.35 J cm⁻³ at 40 kV cm⁻¹ in *Energy Environ. Sci.* **16**, 4511-4521 (2023)). Therefore, our performance comparison and testing methods are not only fair but also align with common practices reported in other studies.

Fig. R3

The P - E loop and energy storage properties for NN-CZ-0.25BNT ceramics.

7. Provide endurance test results to prove reliable energy storage applications.

Reply: Thanks for your good suggestions. Endurance test is an important indicator for measuring the reliable application of energy storage materials. To measure operational reliability, NN-CZ-0.25BNT ($x = 0.25$) ceramic undergoes fatigue testing under electric field cyclic loading (Fig. R4a). It is obvious that the P - E loops always remain a slim shape with small polarization hysteresis during the fatigue testing process. After 10⁶ consecutive electrical cycling tests under an electric field of 60 kV cm⁻¹, the variations of U_e and η for the $x = 0.25$ ceramic are less than 2% and 1% (Fig. R4b), respectively, which demonstrates outstanding fatigue stability. The relevant cycling fatigue experiments and descriptions have been added to the manuscript in red (Page 7; Supplementary Fig. S20, Page S23).

Fig. R4

(a) P - E loops, and (b) U_e and η as a function of cycle number at an electric field of 60 kV mm⁻¹.

Thank you again for your outstanding contribution to the manuscript.

Reviewer #2 (Remarks to the Author):

This manuscript presents a major advance in dielectric energy storage by designing a hierarchical-heterostructure NaNbO_3 -based antiferroelectric ceramic, achieving an ultrahigh recoverable energy density of 21.0 J cm^{-3} at 90% efficiency. The authors leverage multi-scale structural control—including interlocked and fishbone nanodomains and BO_6 tilt modulation—to delay polarization saturation and suppress hysteresis, outperforming existing lead-free bulk ceramics.

- A lead-free bulk ceramic achieving record-high U_e (21.0 J/cm^3) and 90% η , alongside ultrafast discharge (32 ns) and high-power density (460 MW/cm^3), marking a significant improvement over prior systems ($\sim 12 \text{ J/cm}^3$ max for NaNbO_3 -based ceramics).
- Yes, this work is highly significant; it clearly surpasses previous benchmarks (e.g., Qi et al., *Adv. Funct. Mater.*, 2019) in both energy density and efficiency for bulk lead-free ceramics. It introduces a broadly applicable design strategy with potential relevance to other energy storage platforms.
- The manuscript's experimental data robustly supports its conclusions. However, additional evidence such as fatigue endurance, batch reproducibility, and comparative benchmarking would further reinforce key claims.
- No fatal flaws, but gaps remain in scalability discussion, long-term reliability, and modeling of mechanisms. These should be addressed for higher-impact publication but do not prohibit publication.
- Yes, the experimental approach—including TEM, P - E loops, dielectric and breakdown testing—is sound and meets the standards of the field.
- While general processing details are provided, more explicit data on sample-to-sample consistency, synthesis repeatability, and sintering control is needed to confirm reproducibility.

Reply: We highly appreciate the positive and professional comments for the present study. Based on the in-depth insights you provided above and the improvement suggestions below, we have provided a detailed response to the superiority and repeatability of performance and structure, and strengthened the discussion section of the article.

The P - E loop of another NN-CZ-0.25BNT ceramic that is similar to the reported energy storage value in the manuscript is shown in Fig. R3, demonstrating high repeatability.

Fig. R3

The *P-E* loop and energy storage properties for NN-CZ-0.25BNT ceramics.

The antiferroelectric nanoclusters with polarization interlocking structure and fishbone polarization configuration shown in Fig. 3 are very interesting local antiferroelectric structure, which are reported in dielectrics for the first time. To dispel the concerns about their universality, we reprocessed the NN-CZ-0.25BNT sample and collected another set of HADDF-TEM images to analyze polarization configuration. As shown in Fig. R5a,b, the polarization interlocking structure with vertically intersecting or embedding polarization nanoclusters with different directions can also be obviously found in region I from another set of HADDF-TEM images. Furthermore, in region II, we can also observe alternating striped polarization regions with periodicities of $n = 2$ along $[001]_c$ and short-range alternating striped polarization regions with multiple periodicities along $[010]_c$, which shows fishbone polarization structure (Fig. R5c,d). Clearly, the similar antiferroelectric polarization configurations of polarization interlocking structure and fishbone polarization structure can be detected from another set of HADDF-TEM images (Fig. 3), demonstrating that the novel configurations are widely present in NN-CZ-0.25BNT ceramics in this work.

Figure R5

Local polarization structure for another set of HAADF-TEM images. Atomic-resolution HAADF image of the $x = 0.25$ ceramic with the corresponding (a) cation displacement vectors showing polarization interlocking structure and (b) two-dimensional contours of polarization angles for region I. Atomic-resolution HAADF image of the $x = 0.25$ ceramic with the corresponding (c) cation displacement vectors showing fishbone polarization configuration and (d) two-dimensional contours of polarization angles for region II.

The detailed responses to other specific comments are as follows:

Comments and Scope for Improvement

1. Add a summary table comparing U_e , η , E_b , and $t_{0.9}$ with prior top-performing lead-free systems.

Reply: Thanks for your good suggestions. The summary table comparing U_e , η , E_b , and $t_{0.9}$ with prior top-performing lead-free systems including NaNbO_3 , BaTiO_3 , BiFeO_3 , $\text{Bi}_{0.5}\text{K}_{0.5}\text{TiO}_3$, $\text{Bi}_{0.5}\text{Na}_{0.5}\text{TiO}_3$ -based ceramics have been supplemented in the Supplementary Information in red (Page S26, Supplementary Table S2; Page7). It is worth noting that the compared ceramic properties are all from recently published high-level journals (*Acta Mater.* **278**, 120278 (2024); *Adv. Mater.* **36**, 2313285 (2024);

Adv. Mater. **34**, 2204356 (2022); *Acta Mater.* **240**, 118286 (2022); *Nat. Commun.* **15**, 6754 (2024); *J. Am. Chem. Soc.* **146**, 460-467 (2023); *Nat. Commun.* **16**, 807 (2025); *J. Am. Chem. Soc.* **146**, 3498-3507 (2024); *Energy Storage Mater.* **43**, 383-390 (2021)), representing the widely accepted level of high performance. In addition, these do not include single-layer thick films and multilayer ceramic capacitors prepared using casting technology, which cannot be directly compared with ceramics. Clearly, the overall energy storage properties of the prepared NN-CZ-0.25BNT ceramic in our work exhibit significant superiority over other top-performing lead-free samples, even comparable to advanced multilayer ceramic capacitors (*Science* **384**, 185-189 (2024) ($U_e \sim 20.8 \text{ J cm}^{-3}$); *Nat. Mater.* **19**, 999-1005 (2020) ($U_e \sim 21.5 \text{ J cm}^{-3}$); *Nat. Commun.* **14**, 1166 (2023) ($U_e \sim 14.0 \text{ J cm}^{-3}$)).

Supplementary Table S2

A comparison of the comprehensive performance between NN-CZ-0.25BNT ceramic and other representative lead-free energy storage ceramics.

Compositions	U_e (J cm^{-3})	η (%)	E_b (kV mm^{-1})	$t_{0.9}$ (ns)	Ref.
0.88(0.94NaNbO ₃ -0.06BiFeO ₃)- 0.12Sr _{0.7} Bi _{0.2} □ _{0.1} Ti _{0.75} Ta _{0.2} □ _{0.05} O ₃	12.25	83	78	88	[99]
0.85BaTiO ₃ -0.15(Bi _{0.5} Na _{0.5})(Zn _{1/3} Nb _{2/3})O ₃	11.6	96.1	58	41	[14]
0.848(Na _{0.52} K _{0.48})(Sb _{0.035} Nb _{0.965})O ₃ - 0.012SrZrO ₃ -0.14(Bi _{0.5} Na _{0.5})ZrO ₃	13.1	90	74	35.2	[100]
0.62Bi _{0.9} La _{0.1} FeO ₃ -0.3Ba _{0.7} Sr _{0.3} TiO ₃ - 0.08NaNb _{0.85} Ta _{0.15} O ₃	15.9	87.7		5700	[101]
0.7Bi _{0.47} Na _{0.47} Ba _{0.06} TiO ₃ - 0.3Sr _{0.7} La _{0.2} Ta _{0.2} Ti _{0.75} O ₃	15.48	90.02	71	33	[102]
Bi _{0.25} Na _{0.25} Ba _{0.5} Ti _{0.92} Hf _{0.08} O ₃	16.21	90.5	80	30.6	[103]
Bi _{0.2} Na _{0.2} K _{0.2} La _{0.2} Sr _{0.2} (Ti _{0.95} Nb _{0.05})O ₃	16.4	90	85	20	[104]
(Bi _{0.35} K _{0.175} Ba _{0.3} Na _{0.175})(Ti _{0.7} Zr _{0.3})O ₃	17.3	88.5	78	30	[105]
0.90NaNbO ₃ -0.10BiFeO ₃	18.5	78.7	91	14	[12]
NN-CZ-0.25BNT	21.0	90	90	32	Our work

2. Explicitly quantify improvement over established NaNbO₃-based ceramics and thin-film benchmarks.

Reply: Thanks for your good comments. Bulk ceramics (generally 5-15 J cm^{-3}), multilayer ceramic capacitors (generally 10-25 J cm^{-3}), and thin films (generally 80-150 J cm^{-3}) are three important types of energy storage dielectrics, and their energy densities are not on the same order of magnitude. Thus, the energy storage performance of ceramics cannot be directly compared to thin films, as the latter have higher preparation quality and smaller thickness and electrode area (*Prog. Mater. Sci.* **102**, 72-108 (2019); *Nature* **637**, 1104 (2025); *Science* **384**, 185-189 (2024); *Nat. Commun.* **16**, 807 (2025); *J. Appl.*

Phys. **133**, 110904 (2023)). It is known that the previous benchmark for the energy storage properties ($U_e \sim 12.2 \text{ J cm}^{-3}$ and $\eta \sim 69\%$) of NaNbO_3 -based ceramics has been reported in 2019 (*Adv. Funct. Mater.* **29**, 1903877 (2019)), exhibiting large progress in lead-free bulk ceramics. Subsequently, in 2021, Jiang et al. reported ultrahigh energy storage density and enhanced efficiency ($U_e \sim 18.5 \text{ J cm}^{-3}$ and $\eta \sim 78.7\%$) in NaNbO_3 -based antiferroelectric ceramics (*Energy Storage Mater.* **43**, 383-390 (2021)), making the highest energy storage density among lead-free energy storage ceramics at that time. However, they have all exposed the challenge of achieving both ultrahigh energy storage density and ultrahigh efficiency ($\eta < 80\%$), and in the following years, the performance development of bulk ceramics has been stuck in a dilemma where comprehensive enhancement proves difficult.

In this work, the U_e of 21.0 J cm^{-3} is realized in NN-CZ-0.25BNT lead-free antiferroelectric ceramic, which shows a huge breakthrough ($U_e \geq 20 \text{ J cm}^{-3}$) in performance compared to the above reported NaNbO_3 -based energy storage bulk ceramics, especially with ultrahigh efficiency ($\eta \geq 90\%$). The bottleneck of simultaneously achieving ultrahigh energy density ($U_e \geq 20 \text{ J cm}^{-3}$) with ultrahigh efficiency ($\eta \geq 90\%$) in bulk ceramics has been overcome in this work by designing hierarchical heterostructures in antiferroelectric ceramics. To quantitatively evaluate the trade-off between U_e and η , the figure of merit $U_F = U_e / (1 - \eta)$ is also applied to express the comprehensive energy storage properties (*J. Materiomics* **11**, 100980, (2025)). Clearly, the $x = 0.25$ ceramic exhibits the highest U_F value of 210 J cm^{-3} among NN-based energy storage ceramics, indicating the optimal overall energy storage performance (Fig. R6). The descriptions of the benchmark performance for the established NaNbO_3 -based ceramics and improvements have been added to the Supplementary Information in red (Page S15, Supplementary Fig. S13). Thank again.

Fig. R6

Comparisons of U_F between our ceramics and other representative NN-based energy storage ceramics.

3. Add charge-discharge cycling data (e.g., 10^4 – 10^6 cycles).

Reply: Thanks for your good suggestions. To measure cycling reliability, NN-CZ-0.25BNT ($x = 0.25$) ceramic undergoes fatigue testing under electric field cyclic loading (Fig. R4a). It is obvious that the P - E loops always remain a slim shape with small polarization hysteresis during the fatigue testing process. After 10^6 consecutive electrical cycling tests under an electric field of 60 kV cm^{-1} , the variations of U_e and η for the $x = 0.25$ ceramic are less than 2% and 1% (Fig. R4b), respectively, which demonstrates outstanding fatigue stability. The relevant cycling fatigue experiments and descriptions have been added to the manuscript in red (Page 7; Page S23, Supplementary Fig. S20).

Fig. R4

(a) P - E loops, and (b) U_e and η as a function of cycle number at an electric field of 60 kV mm^{-1} .

4. Support the proposed polarization unlocking and restoring force mechanisms with basic phase-field simulations or schematic energy landscapes.

Reply: Thanks for your good comments. Providing phase field simulations and energy landscape diagrams can indeed demonstrate the mechanism of structural enhancement from another perspective. However, phase field simulation cannot simulate based on the characteristics of real components, and can only consider the structure itself for theoretical simulation, lacking authenticity. The schematic diagram of energy landscape also has the problem of lack of basis (*Adv. Mater.* **34**, 2205787 (2022)). Here, our proposed polarization unlocking and restoring force mechanisms can be clearly confirmed by in situ TEM during electric field loading and unloading processes (Supplementary Fig. S16), providing the most direct experimental evidence. The detailed explanation has been elaborated in the Supplementary Information (Page S18-19). Moreover, the anomalous increase in polarization with increasing electric field should be attributed to the gradual release of polarization interlock driven by high electric field, which can also indirectly indicate polarization unlocking (Supplementary Fig. S15).

Notably, machine learning force field assisted molecular dynamics simulation is an effective method of revealing the internal structure and its evolution under external fields with an accuracy close to that of density functional theory calculations. It considers the real composition and has high reliability, but the implementation process is rather difficult and challenging. In future research, we will strive to use molecular dynamics simulations to illustrate structural issues and enhancement mechanisms. Thanks again.

5. In discussion, acknowledge that thin films (e.g., Pan et al., *Science*, 2019) can achieve higher U_e via higher E_b . Position the work as leading within bulk ceramics, with paths to bridge toward film-like performance.

Reply: Thanks for your good suggestions. Compared to energy storage ceramics, it is widely recognized that higher U_e can be universally obtained in thin films through achieved higher E_b (*Science* **365**, 578-582 (2019)). The $x = 0.25$ ceramic exhibits outstanding breakdown resistance strength and a larger E_b than that of the most reported energy storage ceramics (*Nat. Commun.* **13**, 3089 (2022); *J. Am. Chem. Soc.* **145**, 11764-11772 (2023)), which can build a bridge to achieve high energy storage properties closer to thin films. The relevant revisions have been added to the manuscript in red (Page 6).

6. Include permittivity vs. E curves in supplementary.

Reply: Thanks for your good comments. The electric field-dependent dielectric constant for $x = 0.25$ ceramic has been supplemented in the Supplementary Information (the inset in Supplementary Fig. S4e). Due to this testing electric fields are higher than that of the temperature-dependent dielectric spectra, the size of the test samples is consistent with the energy storage testing. As shown in Fig. R7, as the electric field increases, the dielectric constant shows a decreasing trend, which is similar to other reported relaxors.

Fig. R7

The electric field-dependent dielectric constant for $x = 0.25$ ceramic measured at 10 Hz.

7. Simplify overly long sentences. Add subheadings in the Results section for readability. Define terms like “interlocking” and “fishbone polarization” when first used.

Reply: Thanks for your good suggestions. In order to enhance the readability and standardization of the manuscript, we have optimized some excessively long sentences to make them simple and easy to understand. The subheadings have been supplemented and improved in the Results and discussion section (Page 3). These two configurations of “polarization interlocking structure” and “fishbone polarization configuration” are defined based on the distribution characteristics and morphological features of their polarization vectors. In region I, these polarization nanoclusters are vertically interlaced and intertwined with each other, exhibiting a structural feature analogous to mutual interlocking. Thus, we term this configuration a “polarization interlocking structure”. In region II, the alternating striped polarization regions with periodicities of $n = 2$ along $[001]_c$ and short-range alternating striped polarization regions with multiple periodicities along $[010]_c$ can be verified. These slender striped polarization regions () are widely distributed, exhibiting a morphological feature analogous to fishbones, which can be well defined as “fishbone polarization structure”. Based on your suggestions, we have supplemented the above content in the manuscript in red to highlight the characteristics and definitions of polarization interlocking structure and fishbone polarization structure (Page 4-5; Page S9-S10).

Thank you again for your outstanding contribution to the manuscript.

Reviewer #3 (Remarks to the Author):

The authors demonstrate the feasibility of enhancing energy storage performance by designing the composition NN-CZ-xBNT, synthesized via a conventional solid-state route. The topic of the manuscript is interesting and timely. However, in the current format, the manuscript faces a lot of issues such as structural imbalance and in-depth explanation. This is mandatory for a reputed journal like Nat. comm. Therefore, the manuscript requires a major revision, and the following points must be addressed before considering for publication.

Reply: We highly appreciate the positive and professional comments for the present study.

Comments

1. As we are moving towards the miniaturized electronics world, the authors are expected give the importance of ceramics capacitors over the thin film capacitors in application point of view in the introduction.

Reply: Thanks for your good suggestions. From a practical application standpoint, compared with thin film capacitors, ceramic capacitors offer distinct advantages, including simple fabrication, low cost, high power and energy storage capacity, thus attracting growing attention in the field of dielectric energy storage. The ceramic capacitors exhibit important application value in pulse power devices. The relevant revisions have been added to the manuscript in red (Page 2).

2. The authors mentioned “Thus, tailoring the polarization behavior of RFE ceramics is crucial to achieving high energy storage performance (ESP) and meeting the growing demands of advanced capacitor technologies”. It is anticipated to give why RFE ceramics are needed for advanced capacitor technology.

Reply: Thanks for your comments. Relaxor ferroelectric ceramics can simultaneously achieve high P_m , low P_r , high E_b , and excellent stability, which are expected to achieve high energy storage density and efficiency (*Science* **384**, 185-189 (2024); *Adv. Mater.* **34**, 2204356 (2022)). Relaxor ferroelectric ceramics play an important role in the field of dielectric energy storage and are widely studied and reported systems. However, there are no relevant statements of “Thus, tailoring the polarization behavior of RFE ceramics is crucial to achieving high energy storage performance (ESP) and meeting the growing demands of advanced capacitor technologies” in the manuscript and they are not applicable to this manuscript. Thank again.

3. Still, it is unclear in the introduction section that how BNKT-NN28 ceramic capacitors compete with MLCCs and thin films. Please go through the following work [<https://doi.org/10.1016/j.jmat.2024.100980>]. Also, I suggest the authors to make a comparison table of U_e and U_f with other BNT ceramic based systems in the final part of result and discussion.

Reply: Thanks for your comments. We did not report the component of BNKT-NN28 and the relevant statements in the manuscript. Based on question 2 and 3, there may have been some misunderstandings or incorrect operations. It is known that bulk ceramics (generally 5-15 J cm⁻³), multilayer ceramic capacitors MLCCs (generally 10-25 J cm⁻³), and thin films (generally 80-150 J cm⁻³) are three important types of energy storage dielectrics, and their energy densities are not on the same order of magnitude.

Thus, the energy storage performance of ceramics cannot be directly compared to MLCCs and thin films, as the two have higher preparation quality and smaller thickness (*Prog. Mater. Sci.* **102**, 72-108 (2019); *Nature* **637**, 1104 (2025); *Science* **384**, 185-189 (2024); *Nat. Commun.* **16**, 807 (2025); *J. Appl. Phys.* **133**, 110904 (2023)).

Although this comment is not relevant to the content of this manuscript, the suggestion of using U_F to evaluate the comprehensive energy storage performance is worth adopting. To quantitatively evaluate the trade-off between U_e and η , the figure of merit $U_F = U_e / (1 - \eta)$ is applied to express the comprehensive energy storage properties (*J. Materiomics* **11**, 100980, (2025)). Clearly, the $x = 0.25$ ceramic exhibits the highest U_F value of 210 J cm⁻³ among NN-based energy storage ceramics, indicating the optimal overall energy storage performance (Fig. R6). The relevant content has been supplemented and the literature has also been reasonably cited in the manuscript in red (Page 5-6, Page S15, Supplementary Fig. S13b). Thank again.

Fig. R6

Comparisons of U_F between our ceramics and other representative NN-based energy storage ceramics.

4. I suggest the authors to follow the section division as per journal guideline. For instance, it is given subheading likes “General structure characterizations” and “Hierarchical heterostructure and BO6 tilts”, which feels inappropriate. So, please check it and make it more simple headings.

Reply: Thanks for your good suggestions. According to the journal regulations, we have added topics for Results and discussion. Based on this, we have set subheadings “General structure features” and “Hierarchical heterostructures” to modify and simplify “General structure characteristics” and “Hierarchical heterostructures and BO₆ tilts”, respectively. The subheading “Local polarization configuration” has been deleted. The relevant content has been revised in the manuscript in red (Page 3-4).

5. The authors proposed a hierarchical heterostructure. However, the work NN-CZ-xBNT suggest doping. The authors need to give a proper explanation why they chose the term heterostructure. Also, in Fig. 1, the mechanism of forming nanoclusters featuring, the so-called interlocked polarization structure and fishbone polarization configuration is unclear. For instance, it is unclear how the ordered BO_6 tilts forms nanocluster. In the supplementary Fig. S2 also, it is not clear. The authors need to say how CZ and BNT transforms the macrodomains to these mentioned nanoclusters. Instead of simply providing low- t and non-polar. Otherwise, the authors meant PNR, it is agreeable. If it is not, the authors need to give literature or theoretical evidence to prove these terms, especially, fishbone polarization configuration. This is the core part of the original article and should be very clear to the audience. Therefore Fig. 1 must be reader friendly with precise explanation. This is lacking here.

Reply: Thanks for your good comments. We are pleased to engage in in-depth discussions with you regarding the design concept of this manuscript. In response to your concerns, we have prepared the following detailed responses:

Firstly, hierarchical heterostructures are constructed by finely tuning polarization and oxygen octahedral (BO_6) tilt at two local scales. Notably, structural heterogeneity at the local scale further induces differences at the mesoscopic scale (domain) and microscopic scale (grain), ultimately leading to a multi-scale heterostructure spanning the local, mesoscopic, and microscopic levels. For this reason, we define this multi-scale heterostructure as “hierarchical heterostructures”.

Secondly, structural design is closely related to component selection. Our prior work has involved systematic studies and in-depth understanding of NN-based antiferroelectric systems (*Acta Mater.* **208**, 116710 (2021); *Adv. Funct. Mater.* **29**, 1903877 (2019); *Mater. Today* **60**, 1369-7021 (2022); *Adv. Mater.* **34**, 2205787 (2022); *Nat. Commun.* **16**, 886 (2025)). We propose that the heterogeneity of two local-scale structural features—polarization and BO_6 tilt—requires contradictory component design by introducing components with different functional regulatory properties. For example, introducing components with low tolerance factor (t) stabilizes the antiferroelectric phase and enhances BO_6 octahedral tilt, while high- t components weaken BO_6 octahedral tilt (*Mater. Today* **60**, 1369-7021 (2022); *Dalton Trans.*, **44**, 10763 (2015); *Acta Mater.* **208**, 116710 (2021)). Given the need to maintain a stable antiferroelectric phase, we prioritize components with $t < 1$. In terms of polarization regulation, it is recognized that non-polar components reduce the overall polarization of materials, whereas highly polar

components enhance local polarization. Therefore, we selected CZ and BNT as two regulatory components for NN-based relaxor antiferroelectric ceramics, which can induce heterogeneity in local polarization and BO_6 tilt. It is precisely this “contradictory design” that enables the formation of hierarchical heterostructures.

Thirdly, according to the established theory of relaxor ferroelectrics (*J. Mater. Sci.* **41**, 31-52 (2006)), the introduction of multiple elements (which occupy the same crystallographic sites) strengthens the random fields, thereby disrupting the long-range ordered antiferroelectric structure. The addition of CZ and BNT can transform the long-range domain structure and ordered BO_6 tilt structure into short-range ordered features (e.g., antiferroelectric nanoclusters) or disordered BO_6 tilt distributions. This phenomenon has been strongly corroborated by our previous studies (*Adv. Funct. Mater.* **29**, 1903877 (2019); *Mater. Today* **60**, 1369-7021 (2022)). Importantly, the degree of disruption to the long-range ordered structure depends on the number of introduced components.

Fourthly, the formation of local nanocluster/nanoregion structures within ordered BO_6 tilt regions has also been validated in our prior work (*Mater. Today* **60**, 1369-7021 (2022)). This is feasible. It is worth noting that in previous reports, nanoclusters or nanoregions exhibit ferroelectric nature rather than antiferroelectricity in this work.

Fifthly, it is important to clarify that this structural design describes a qualitative transition (from structural order to disorder) rather than a quantitative model. We cannot precisely model the “polarization interlocking structure” or “fishbone polarization structure” in detail; instead, these two configurations correspond to a transition process: from an ordered state (antipolar configuration) to a fully disordered state (relaxor ferroelectric configuration: randomly oriented polymorphic PNRs). Both configurations were first identified and verified via atomic-resolution HAADF-STEM characterizations. As reflected in the manuscript, the polarization interlocking structure exhibits higher antiferroelectric ordering compared to the fishbone polarization structure—this difference corresponds to their respective ordered and disordered BO_6 tilt configurations. These two configurations are defined based on the distribution characteristics and morphological features of their polarization vectors; to our knowledge, this is the first time such local polarization configurations of relaxor antiferroelectrics have been reported, with no prior references or theoretical precedents. Notably, these configurations cannot be categorized as polar nanoregions (PNRs) of relaxor ferroelectrics; instead, we term them “antiferroelectric nanoclusters”. Given the unpredictability of local polarization configurations, we cannot provide the

specific structure of antiferroelectric nanoclusters in Fig. 1. Instead, Fig. 1 only illustrates the transition process from long-range order to short-range order or disorder. We have carefully considered the schematic diagram of Fig. 1 before.

Finally, following your suggestion, we have revised Fig. 1 to eliminate potential misunderstandings regarding the design concept and content. We effectively merged and integrated the component design strategy for hierarchical heterostructures (originally presented in Fig. S2) into the revised Fig. 1 (Fig. R8). This revision can clarify the core ideas and themes of the manuscript, thereby improving the readability of the work for readers. The relevant revisions have been added to the manuscript **in red** (Page 3; Page 11, Fig. 1). Thank you again for your valuable insights and in-depth discussions.

Fig. R8

Schematic diagram of designing hierarchical heterostructures for enhancing energy storage properties. **a**, Component design strategy for designing hierarchical heterostructures. **b**, Designing hierarchical heterostructures to optimize polarization evolution paths in NaNbO_3 -based lead-free antiferroelectrics.

6. However, the work ref [30] cited by the reviewers calculated the γ value for confirming the relaxor nature.

I suggest the authors to do the same.

Reply: Thanks for your good comments. The diffusion coefficient γ can be calculated by the following equation:

$$\frac{1}{\varepsilon} - \frac{1}{\varepsilon_m} = \frac{(T - T_m)^\gamma}{C}$$

where C is the Curie constant. As shown in Fig. R9, the γ values of $x = 0$, $x = 0.10$, $x = 0.15$, $x = 0.20$, and $x = 0.25$ ceramics are 1.32, 1.15, 1.29, 1.30 and 1.31, respectively. It should be noted that γ can only describe the dispersion of the phase transition peak. Furthermore, using the γ to measure the degree of relaxation requires determining the T_m of the material for ensuring high reliability (*J. Mater. Sci.* **41**, 31-52 (2006)). Unfortunately, due to the high degree of relaxation, the T_m values of all samples were extremely low, to the point where the true T_m could not be detected under the initial testing conditions of -120 °C (Fig. S4). This initial testing temperature is also basically at the testing limit starting temperature (range from -150 °C to -120 °C) of conventional LCR meter (*Adv. Mater.* **34**, 2204356 (2022); *Chem. Eng. J.* **470**, 144205 (2023)). Therefore, the γ values we obtained are inaccurate, as they use the maximum value around -120 °C as a reference instead of the actual T_m . From the current trend of dielectric properties, it is extremely difficult to measure T_m even if the temperature is lowered to -150 °C. This situation does not exist in the samples from the ref [30], so reliable γ values can be calculated to confirm the relaxation nature.

In this work, dielectric spectra have confirmed that all samples are in a superparamagnetic state and belong to relaxors, which can also be supported by STEM/TEM, XRD and other techniques. Therefore, considering rigor and reliability, we cannot provide γ values to measure the relaxation nature of the samples.

Fig. R9

Plot of $\ln(1/\varepsilon - 1/\varepsilon_m)$ as a function of $\ln(T - T_m)$ according to the modified Curie-Weiss law.

7. The authors stated that “All samples show the superparaelectric state with low loss characteristic, where ultrasmall weakly coupled PNRs exist.” It should be better to correlate how these ultrasmall weakly coupled PNRs enhance the diffused phase transition and relaxation behaviour. Because by seeing the temperature dependent dielectric spectroscopy graph, it is visible that the room temperature permittivity is significantly reduced (almost 2 times). And the P - E loops in the Fig. S12 does not follow this trend. At the low electric fields, also the PNRs nearly maintained almost same saturation polarization compared to $x=0$ composition, indicating a good response of PNRs to electric field at room temperature. Therefore, I suggest the authors to give a proper explanation for this.

Reply: Thanks for your good comments. According to the relevant theory of relaxor ferroelectrics, PNRs in relaxor ferroelectrics are weakly coupled and different forms of PNRs have different T_c (or phase transition temperatures) due to the differences in local component combinations (*J. Mater. Sci.* **41**, 31-52 (2006)). PNRs with low T_c will preferentially undergo phase transition with increasing temperature. Different PNRs undergo phase transitions at their respective T_c , which collectively exhibits diffuse phase transition behavior in macroscopic dielectric spectra. As is well known, weakly coupled PNRs exist in the superparaelectric state. With the increase of BNT content, the disorder of composition and the enhancement of random fields will lead to a decrease in the size and an increase in the types of PNRs, thereby enhancing the diffuse phase transition and relaxation behavior. These can be easily identified through dielectric spectra.

Indeed, the dielectric constant of the $x = 0$ at room temperature is much higher than that of other samples, even nearly twice as high. However, the P - E loops also follow relevant trends rather than being irregular. It is known that the polarization of materials is related to their dielectric constant. As shown in Fig. R10, S11, and S15, we extracted the maximum polarization values of each sample under different electric fields. It can be clearly observed that the maximum polarization ($8.69 \mu\text{C cm}^{-2}$) of the $x = 0$ under low electric field (6 kV mm^{-1}) is 1.81 times that of the $x = 0.25$ ($4.79 \mu\text{C cm}^{-2}$), which is close to the rate of change of room-temperature dielectric constant. In addition, the polarization of $x = 0.10$, $x = 0.15$, $x = 0.20$, and $x = 0.25$ is very similar under different electric fields, which is consistent with the trend of room-temperature dielectric constant (Fig. S4). Due to the fact that the testing electric field for the dielectric constant (500 mV) is much smaller than that of the P - E loop, the electric field dependence of the dielectric constant also needs to be considered. Generally, the dielectric constant decreases with increasing electric field, and this phenomenon is more significant in ferroelectrics than in relaxor

ferroelectrics. In addition, the quality of sample preparation, structural characteristics such as grain size, domains, and local polarization also have a significant impact on the polarization response under an electric field. These can cause deviations between polarization evolution and dielectric constant evolution. It should be noted that the polarization and dielectric constant changes of our samples exhibit good consistency. Thanks again.

Fig. R10

Maximum polarization of NN-CZ-xBNT ceramics as a function of electric field.

8. The authors used the term “fishbone polarization configuration” multiple times. However, I could not find such terms in the cited literatures [34-36]. If the authors are the first to introduce the term, it should be well defined. For example, the authors should clearly define how this fishbone polarization configuration is different from nanodomains like “polymorphic nanodomains”. Because whatever the authors mentioned in the literature are features pointing towards either PNRs or polymorphic nanodomains.

Reply: Thanks for your good comments. Firstly, the cited original literatures [34-36] consistently confirm that traditional nanodomains or PNRs in relaxor ferroelectrics generally exhibit ellipsoidal morphologies within non-polar matrices (*J. Mater. Sci.* **41**, 31-52 (2006); *Phys. Rev. B* **64**, 184112 (2001); *Ferroelectrics* **217**, 327-333 (1998)), rather than defining two antiferroelectric polarization configurations in the present work. Secondly, both configurations were first identified and verified via atomic-resolution HAADF-STEM characterizations. These two configurations are defined based on the distribution characteristics and morphological features of their polarization vectors; to our knowledge, this is the first time such local polarization configurations of relaxor antiferroelectrics have been reported, with no prior references or theoretical precedents. In region I, these polarization nanoclusters are vertically interlaced and intertwined with each other, exhibiting a structural feature analogous to mutual

interlocking. Thus, we term this configuration a “polarization interlocking structure”. In region II, the alternating striped polarization regions with periodicities of $n = 2$ along $[001]_c$ and short-range alternating striped polarization regions with multiple periodicities along $[010]_c$ can be verified. These slender striped polarization regions are widely distributed, exhibiting a morphological feature analogous to fishbones, which can be well defined as “fishbone polarization structure”. Thirdly, these configurations cannot be categorized as PNRs or polymorphic nanodomains of relaxor ferroelectrics; instead, we term them “antiferroelectric nanoclusters”. As shown in Fig. R11 (*Nat. Commun.* **13**, 3089 (2022); *Energy Environ. Sci.* **16**, 4511-4521 (2023)), in relaxor ferroelectrics, the polarization configuration typically exhibits as multiple randomly oriented PNRs embedded within non-polar matrices, where the polarization orientations of adjacent nanoregions tend to align along similar directions. By contrast, in relaxor antiferroelectrics, antiferroelectric nanoclusters display nearly antiparallel polarization orientations in adjacent nanoregions, exhibiting characteristics of antiferroelectric polarization configurations. It is worth noting that due to the introduction of strong random fields formed by multiple components, the polarization vector in relaxor antiferroelectrics is not absolutely anti-parallel. Instead, both the magnitude and direction of polarization fluctuate within a finite range. Moreover, the commensurate modulated structure may also experience disturbances. Thus, the antiferroelectric nanocluster configuration reported herein differs significantly from PNRs or polymorphic nanodomains in traditional relaxor ferroelectrics. Finally, based on your suggestions, we have supplemented the above content in the manuscript **in red** to better distinguish the polarization configurations of relaxor antiferroelectrics and relaxor ferroelectrics, highlighting the characteristics and definitions of polarization interlocking structures and fishbone polarization structures (**Page 4-5; Page S9-S10**).

Figure Redacted

Fig. R11

The morphological features of polymorphic PNRs/nanoclusters found in relaxor ferroelectrics (*Nat. Commun.* **13**, 3089 (2022); *Energy Environ. Sci.* **16**, 4511-4521 (2023)).

9. Finally, the structure of the manuscript should be well-defined and should be written reader friendly. Please follow the journal guidelines.

Reply: Thanks for your good comments. We have made modifications and optimizations to the structure

of the manuscript in accordance with the suggestions and comments of the reviewers and the journal guidelines to enhance readability.

Thank you again for your outstanding contribution to the manuscript.

Dear reviewers:

We highly appreciate the valuable suggestions and comments on our manuscript entitled “**Design of hierarchical-heterostructure antiferroelectrics for ultrahigh capacitive energy storage**” (NCOMMS-25-51936A). The careful revisions have been made accordingly, which were highlighted in red in the revised manuscript. We hope that these changes adequately address the concerns raised. The responses to the comments are listed point-by-point below:

Reviewer #1 (Remarks to the Author):

The manuscript is ready for publication.

Reply: We highly appreciate the positive comments for the present study. Thank you for your outstanding contribution to improving the quality of our manuscript.

Reviewer #2 (Remarks to the Author):

The authors have incorporated my suggestions and given an explanation/rebuttal wherever required. I recommend the manuscript be considered for publication.

Reply: We highly appreciate the positive comments for the present study. Thank you for your outstanding contribution to improving the quality of our manuscript.

Referee #2's comment on the report of Referee #3:

I appreciate the concern of Reviewer #3 in evaluating the manuscript. However, with due respect to Reviewer #3, I find that the concern cannot be considered a sole reason to reject the paper, and the explanation in the comment itself partially answers the question and supports the authors' stand.

Additionally, in response to my comment #5, the author stated, "Compared to energy storage ceramics, it is widely recognised that higher U_e can be universally obtained in thin films through achieving higher E_b (*Science* **365**, 578-582 (2019)). The $x = 0.25$ ceramic exhibits outstanding breakdown resistance strength and a larger E_b than that of the most reported energy storage ceramics (*Nat. Commun.* **13**, 3089 (2022); *J. Am. Chem. Soc.* **145**, 11764-11772 (2023)), which can build a bridge to achieve high energy storage properties closer to thin films. The relevant revisions have

been added to the manuscript in red (Page 6)".

Thus, the motive is clear in the manuscript, and the authors have done sufficient work.

I have no further comment.

In case a revised comment is essential, I would say:

The authors have incorporated my suggestions and given an explanation/rebuttal wherever required. However, there is scope to improve the introduction further by elaborating on the explanation added "From a practical application standpoint, compared with thin film capacitors, ceramic capacitors offer distinct advantages, including simple fabrication, low cost, high power and energy storage capacity, thus attracting growing attention in the field of dielectric energy storage 5,6".

Reply: We once again thank you for your fair and objective evaluation of the manuscript. Your support will enhance our confidence in the manuscript and stimulate further research interest in this field. The statement has been optimized to "From a practical application standpoint, compared with thin film capacitors, ceramic capacitors offer distinct advantages, including simple fabrication, low cost, high power and energy storage capacity under low applied electric fields (E), thus attracting growing attention in the field of dielectric energy storage", which is marked in red in the manuscript (Page 2).

Reviewer #3 (Remarks to the Author):

I appreciate the efforts of the authors. But the authors were unable to answer the comments properly and therefore, I reject the paper. For instance, to my comment 1, the authors mentioned "From a practical application standpoint, compared with thin film capacitors, ceramic capacitors offer distinct advantages, including simple fabrication, low cost, high power and energy storage capacity, thus attracting growing attention in the field of dielectric energy storage." This cannot be agreeable as scientific community is giving importance to thin films for the integration and even trying to develop flexible thin films [<https://doi.org/10.1016/j.pmatsci.2022.101046>, <https://doi.org/10.1016/j.jeurceramsoc.2024.02.016>]. However, the high breakdown strength is a big issue with the thin film. On the other hand, ceramic capacitors produce better ESD at low voltages. This will minimize the cost of insulation technology and miniaturization of the electronic devices. I clearly gave the hint to the authors, and they were unable to implement it

[<https://doi.org/10.1016/j.jmat.2024.100980>]. Also, I suggested the authors to make a comparison table of U_e and U_f with other BNT ceramic based systems in the final part of result and discussion. This is very important as these Figure of parameters determines how efficient is your ceramics. Event I suggested the authors to make a comparison with BNT based systems and the authors are commenting it is inadequate. Also, the authors selected only a very few works only. Thus, I find the authors response very weak and a renowned journal like Nat. Comm. require high quality analysis and unique results to publish.

Reply: We highly appreciate the comments for the present study. We would like to provide the following responses to your comments:

Firstly, during the first round of revision in response to the review comments, we took your feedback seriously and made sufficient revisions. However, we note that Comments 2 and 3 have no relevance or logical connection to this manuscript. Specifically, the statement “The authors mentioned “Thus, tailoring the polarization behavior of RFE ceramics is crucial to achieving high energy storage performance (ESP) and meeting the growing demands of advanced capacitor technologies”.” **mentioned in Comment 2 does not appear in our manuscript, nor does any content related to relaxor ferroelectric (RFE) ceramics.** Additionally, **the BNKT-NN28 composition referenced in Comment 3 is not included in our work**, making it impossible for us to provide more relevant descriptions rather than “weak response”. It is important to clarify that our study focuses on NN-based antiferroelectric ceramics rather than BNT-based ceramics. It is reasonable to restrict performance comparisons to NN-based ceramics rather than expanding to BNT-based ceramics. **We have followed your suggestion to the greatest extent possible by comparing the U_F values of NN-based ceramics (Supplementary Fig. S13b).** For further clarification: in accordance with suggestions from other reviewers, the data we selected **includes performance metrics of state-of-the-art NN-based energy storage ceramics.** We did not include comparisons with all low-performance NN-based ceramics, as such an approach would be redundant and non-intuitive. Furthermore, our manuscript already includes a systematic performance comparison with other ceramic systems (including AN, BF, BT, KNN, and BNT-based ceramics) (Fig. 4b, Supplementary Fig. S13, and Supplementary Table S2), which we believe is sufficiently comprehensive. Regarding “ U_e ”, this manuscript uses the term to refer to recoverable energy storage density—a standardized definition in the field (*Nature* **637**, 1104 (2025); *Science* **369**,81-84 (2020);

Science **384**, 185-189 (2024))—and the U_e values of the ceramics reported herein have been thoroughly compared (Fig. 4b, Supplementary Fig. S13, and Supplementary Table S2). In contrast, the literature [<https://doi.org/10.1016/j.jmat.2024.100980>] cited in your comment uses “ U_E or U_e ” to denote U_{Rec}/E (recoverable energy storage density per electric field), which may lead to confusion for us and readers. As shown in Table R1, we also compared NN-CZ-0.25BNT ceramics with representative high-performance BNT-based ceramics, and the performance we reported still showed significant superiority, meeting the high requirements of the journal.

Table R1. A comparison of the U_e (U_{Rec}/E) and U_F between NN-CZ-0.25BNT ceramic and the representative BNT-based lead-free energy storage ceramics.

Ceramics	U_e (U_{Rec}/E) (J MV ⁻¹ cm ⁻²)	U_F (J cm ⁻³)	Reference
BNT-based	19.3	164.0	Nat. Commun. 16 , 807 (2025)
BNT-based	20.3	170.6	J. Am. Chem. Soc. 146 , 460-467 (2024)
BNT-based	21.8	155.1	Nat. Commun. 15 , 6754 (2024)
BNT-based	20.8	168.9	J. Am. Chem. Soc. 145 , 19396-19404 (2023)
NN-based	23.3	210	Our work

Secondly, ceramic capacitors and thin film capacitors represent two major categories of widely studied dielectric energy storage capacitors. As per your suggestion, we have already highlighted the advantages of ceramic capacitors over thin film capacitors from an application perspective, which are widely recognized in the dielectric energy storage field. Ceramic capacitors (including multilayer ceramic capacitors, MLCCs) are currently the most widely used and reliable dielectric energy storage devices, and they have also received extensive attention and research in the scientific community. While thin film capacitors are also a focus of scientific research efforts, this fact cannot logically serve as a reason to deny our descriptions of ceramic capacitors.

Thirdly, it is generally recognized in the field of dielectric energy storage that: For thin film capacitors: The film thickness typically ranges from 100 to 500 nm, with a breakdown electric field (E_b) of 1-7 MV·cm⁻¹, a voltage range of 30-300 V, and an energy storage density (ESD or U_e) of 50-200 J·cm⁻³. For ceramic capacitors: The thickness usually varies from 10 to 200 μm, with an E_b of

0.3-1.3 MV·cm⁻¹/30-130 kV mm⁻¹, a voltage range of 1000-5000 V, and an energy storage density of 5-20 J·cm⁻³. Thus, thin film capacitors exhibit higher energy storage density under **low-voltage** conditions—a well-recognized advantage of this category. From the perspective of electric fields, ceramic capacitors generally possess superior energy storage densities at **low electric fields** compared to thin films (*J. Materiomics* **11**, 100980 (2025)). However, due to their smaller volume, their energy storage capacity (as distinct from energy storage density) is lower than that of ceramic capacitors. The ultrahigh E_b of thin film capacitors enables them to achieve high energy storage performance within an extremely small size. Therefore, the statement has been optimized to “From a practical application standpoint, compared with thin film capacitors, ceramic capacitors offer distinct advantages, including simple fabrication, low cost, high power and energy storage capacity under low applied electric fields (E), thus attracting growing attention in the field of dielectric energy storage”, which is marked **in red** in the manuscript (Page 2).

Finally, it is worth noting that the aforementioned comments do not impact the innovation, logic, or validity of this manuscript. Once again, we appreciate your understanding and contributions to the review of this manuscript.

Dear reviewers:

We highly appreciate the valuable suggestions and comments on our manuscript entitled “**Design of hierarchical-heterostructure antiferroelectrics for ultrahigh capacitive energy storage**” (NCOMMS-25-51936B). We also sincerely appreciate your recognition of our work. With your help and advice, the quality of the manuscript has been significantly improved. The responses to the comments are listed point-by-point below:

Reviewer #2 (Remarks to the Author):

I recommend considering the manuscript for publication.

Reply: We highly appreciate the positive comments for the present study. Thank you again for your important contribution to improving the quality of the manuscript.